# Differential patch-leaving behavior during probabilistic foraging in humans and gerbils
Lasse Güldener [1] ✉, Parthiban Saravanakumar[2], Max F. K. Happel[2,3,4], Frank W. Ohl[2,3,5], Maike Vollmer [2,6,7] & Stefan Pollmann[1,3]

Foraging confronts animals, including humans, with the need to balance exploration and exploitation: exploiting a resource until it depletes and then deciding when to move to a new location for more resources. Research across various species has identified rules for when to leave a depleting patch, influenced by environmental factors like patch quality. Here we compare human and gerbil patch-leaving behavior through two analogous tasks: a visual search for humans and a physical foraging task for gerbils, both involving patches with randomly varying initial rewards that decreased exponentially. Patch-leaving decisions of humans but not gerbils follow an incremental mechanism based on reward encounters that is considered optimal for maximizing reward yields in variable foraging environments. The two species also differ in their giving-up times, and some human subjects tend to overharvest. However, gerbils and individual humans who do not overharvest are equally sensitive to declining collection rates in accordance with the marginal value theorem. Altogether this study introduces a paradigm for a between-species comparison on how to resolve the exploitation-exploration dilemma.

"Should I stay or should I go? "In natural situations, foraging animals constantly find themselves confronted with the dilemma to either keep exploiting a current source of energy thereby depleting it more and more, or to move on and explore the environment for novel sources of energy. Overall, there is no clear rule for decision-making on when to stop exploiting and when to start exploring. This so-called patch-leaving behavior can be driven by external events as well as by internal urges[1,2]. Here, we report results from a behavioral study in which we tested 52 human participants in a probabilistic foraging task and compared their patch-leaving behavior to that of 18 gerbils. For this purpose, we designed two distinct foraging tasks suitable for the respective species. Yet, the two tasks were similar enough in their principle operationalizations to allow the comparison between human and rodent patch-leaving decisions. Our central goal was to analyze the reward-dependent foraging behavior across the two species using two specific paradigms that can eventually pave the way for future research on cross-species comparisons. Importantly, our approach does not imply that specific patch-leaving behaviors can be expected to occur universally, independent of the experimental paradigms used.

The human participants engaged in a visual foraging task embedded within a visual search paradigm[3,4]. This task mirrors the foraging environments encountered by animals, including our hunter-gatherer ancestors, where resources are spatially and temporally distributed across patches, such as different forest districts with varying prey richness. Foraging in such environments involves serial decision-making and incurs temporal travel costs as animals move from one patch to another. This aspect of foraging is better captured in a serial visual search task, as opposed to traditional bandit-like gambling tasks used to study the exploration-exploitation dilemma[5,6], where decisions involve simultaneous choices. In our task, participants searched for target items among distractors on monitor displays, deciding whether to continue searching the current display or switch to a new one by pressing a button. Each successful find earned a monetary reward. This approach aligns with previous studies combining serial visual search with patch-based foraging tasks[3,4]. Adapting similar experimental conditions for rodents, we used an established paradigm[7] where gerbils foraged in a box-like arena with two spouts dispensing food rewards. Both human and rodent tasks shared a probabilistic and patch-based structure, with reward probabilities decreasing exponentially, thus simulating a natural foraging

[1]Department of Experimental Psychology, Institute of Psychology, Otto-von-Guericke-University, 39106 Magdeburg, Germany. [2]Department of Systems Physiology of Learning, Leibniz-Institute for Neurobiology, 39118 Magdeburg, Germany. [3]Center for Behavioral Brain Sciences, 39106 Magdeburg, Germany. [4]Medical School Berlin, Faculty for Medicine, 14197 Berlin, Germany. [5]Institute of Biology, Otto-von-Guericke-University, 39106 Magdeburg, Germany. [6]Department of Experimental Audiology, Otto-von-Guericke-University, 39120 Magdeburg, Germany. [7]University Clinic of Otolaryngology, Head and Neck Surgery, Otto-von-Guericke-University, 39120 Magdeburg, Germany. ✉e-mail: lasse.gueldener@ovgu.de

environment where success declines the longer foragers remain in the same patch. Thus, both animal and human subjects were comparably forced to make patch-leaving decisions to achieve optimal foraging.

A plethora of theories and formal models have been proposed to model the decision-making process that results in exploration and to predict the optimal time for patch-leaving[8–10]. One common assumption is that variations in environmental reward probabilities and patch quality influence patch-leaving behavior, helping to identify decision rules about the optimal time to leave a patch[11,12]. Our study focuses on the extent to which human and rodent foragers utilize implicit probabilistic knowledge and their own reward histories in making patch-leaving decisions. The Marginal Value Theorem (MVT), introduced by Charnov in 1976[8], is a prominent model in our research. It suggests that foragers should leave a patch when the instantaneous collection rate (ICR) of rewards falls below the mean collection rate (MCR) of the environment, implying that foragers must access and utilize their memory of recent foraging experiences to track their current and average energy intake. To promote the use of a probabilistic patch-leaving rule based on reward history, we manipulated reward probabilities in two ways. Firstly, both experimental paradigms introduced an exponential decay of rewards, mimicking a quickly depleting food source[7]. In the rodent paradigm, after a gerbil received a reward from nose-poking, the probability of receiving another reward at the same spout decreased exponentially toward zero. Similarly, in the human task, once a target item was collected, it became inactive and remained on the display; additionally, other target items were progressively deactivated. This meant that fixating on these deactivated targets did not yield a reward. The number of target deactivations followed the same exponential decay functions used in the

gerbil task. This setup increased the difficulty of the search over time, encouraging participants to switch to a new display. At the same time, we avoided using a Poisson distribution for reward probabilities. Under a Poisson distribution, the number of prey in a patch is expected to be random and independent of the time spent in the patch[13]. Consequently, the expected rate of reward does not decline over time, as it typically would in environments where resources deplete due to time spent in the patch. With a Poisson reward distribution, the optimal strategy is to spend a fixed amount of time in each patch, regardless of the number of rewards collected, and then move to the next patch. This 'fixed-T rule' maximizes the expected rate of reward per unit time, as the average number of reward items found in a given area or time is constant and does not depend on the duration of the search[13]. This strategy does not require knowledge of the subjective average and instantaneous reward intake rates. However, this becomes suboptimal when the reward probability decreases as a function of residence time, as is the case in our tasks.

In addition, the initial reward probabilities varied randomly among high (100%), medium (75%), and low (50%) levels. Importantly, these patch qualities were not initially indicated to the participants. This design prevented patches from having a uniform number of available rewards, which would make the optimal patch-leaving rule a simple matter of collecting a fixed number of rewards, such as 10, before leaving (i.e., the fixed-N rule, see Fig. 1b). Such a strategy becomes suboptimal in environments where patch quality varies greatly, and foraging time is limited[12]. In low-quality patches, foragers adhering to the fixed-N rule would spend excessive time collecting the predetermined number of rewards, reducing their overall capture rate given the limited available foraging time. Similarly, using a fixed-T rule in

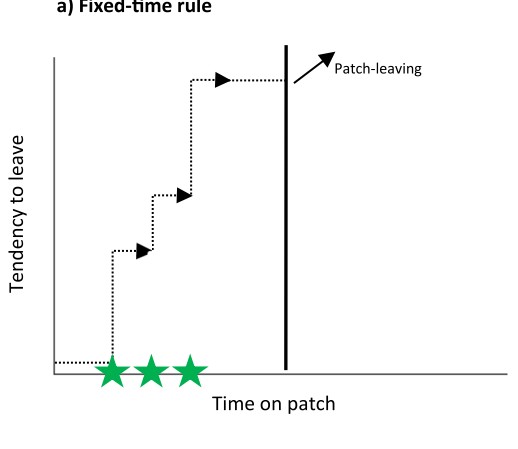

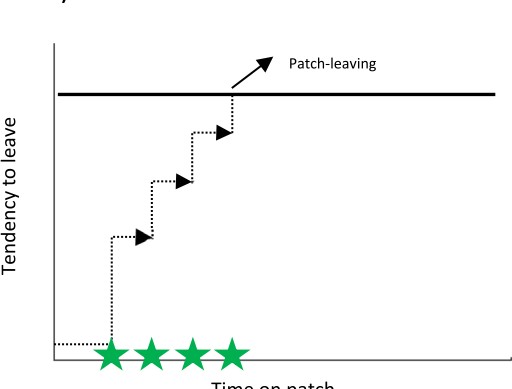

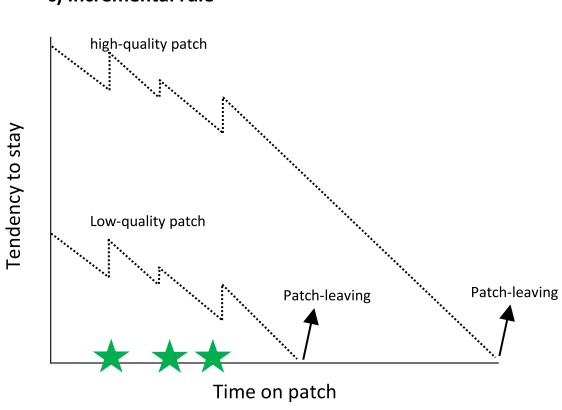

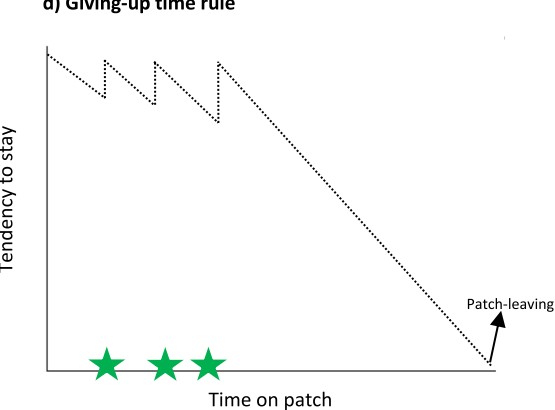

**Fig. 1 | Simple heuristics to time patch-leaving decisions.** Using a fixed-time rule (**a**), the patch is left independent of the number of prey encounters (green stars), whereas a patch is left after a fixed number of prey encounters have been found if a fixed-number rule is used (**b**). According to incremental rule (**c**), each prey capture increases the probability of staying in a patch, postponing the patch-leaving. Using the giving-up-time rule (**d**), the tendency to stay in the patch declines as a function of an unsuccessful search and each prey capture resets this tendency to a maximum.

poor-quality patches would result in fixed time intervals of foraging with few or no rewards, as foragers would continue until the predetermined time is reached, regardless of success. Therefore, environments with highly variable patch qualities necessitate a different behavioral adaptation for optimal foraging.

In such environments, it is challenging to reliably estimate the quality of a patch upon entry, but each reward capture suggests that the current patch is of high quality, thereby increasing the tendency to stay[12]. This results in an incremental mechanism (see Fig. 1c). The probability of staying in a patch initially decreases upon entering a new patch, but each reward capture subsequently increases the likelihood of remaining[14]. This rule allows foragers to rely primarily on their success in foraging to guess the patch quality, rather than needing to estimate it initially, which can be difficult or impossible. Foragers using this rule tend to spend more time in high-quality patches compared to medium and low-quality ones. Additionally, each new reward capture should incrementally extend the forager's residence time, regardless of the patch's quality. Therefore, regressing residence times on the number of rewards captured at the subject level should yield positive slopes.

Like the incremental rule, the Giving-Up Time (GUT) rule[10,15] (see Fig. 1d) does not require prior knowledge or judgment about the quality of a patch. The GUT rule posits that a forager tolerates only a certain amount of time without finding a new reward since the last capture. Once this temporal threshold is exceeded, the forager leaves the patch. This mechanism can be compared to a countdown timer that starts as soon as the forager enters a patch. Each reward capture resets and restarts this timer. If no reward is captured before the timer expires, the forager departs from the patch. Consequently, in patches with high reward probability, where prey is encountered more frequently, the timer is reset more often, leading foragers using the GUT rule to spend more time in high-quality patches compared to lower-quality ones. GUT durations should also be consistent within individuals and exceed the durations of intervals between two target captures (inter-target times, ITT): if a subject's GUT threshold is 4 s, then their ITT should always be shorter than or equal to 4 s because the subject will leave the patch once the ITT exceeds this threshold. Like the incremental rule, the GUT rule utilizes past success to estimate future success and does not necessitate a prior assessment of patch quality. This makes the GUT rule particularly effective in environments with patches of varying quality and where patch quality is difficult to determine in advance[12].

Given the rapid depletion of rewards and the unpredictable variations in patch quality, we hypothesized that both humans and gerbils are sensitive to variable reward probabilities and adopt either an incremental or a GUT rule to optimize their patch-leaving decisions, similar to mice tested in a comparable foraging task whos' patch-leaving behavior was best explained by a model where reward captures incrementally increased the probability of staying, analogous to the incremental rule. This led to optimal timing of patch-leaving as the mice's ICRs at the time of leaving were statistically indistinguishable from the MCR[7]. Thus, we predicted that also our gerbils' ICRs would approximate their MCRs at the time of leaving.

Previous studies have shown that humans performing a visual search-based foraging task do not necessarily conform to this prediction of the MVT, often residing longer than predicted by the theory. This appears to be particularly the case when subjects have the option to switch between target types within the same display, when the patch quality varies greatly[4] (i.e., when subjects foraged in patches that had one out of ten randomly chosen reward probabilities), or when visual information is reduced to the extent that foragers can no longer discern whether a target item is associated with a reward[4]. This evidence suggests that when foraging tasks become more complex, patch-leaving behavior may no longer align with the MVT. Compared to these studies, our human task was less complex because subjects searched for one target type only (no switches between targets), and the underlying reward probability was varied in only three conditions. Given these simpler conditions, we also expected humans' ICRs to approximate the MCRs at the time of leaving, consistent with the MVT.

## Results

### The number of reward captures and residence times increased with patch quality

Timing patch-leaving decisions by a fixed number of reward capture (i.e., the fixed-n rule) would result in equal numbers of rewards- across patches. Inconsistent with this, our human foragers showed an increased number of reward captures with increasing patch quality, $F(2,82) = 449.848$. The average number of reward captures did not differ between humans tested in the PC laboratory and those tested in the fMRI lab, $p = 0.259$. The highest number of rewards was yielded in high-quality patches [M = 11 ± 3], a lower number of rewards was captured in medium-quality patches [M = 8 ± 3, $t(40) = -4.606$, $p = 0.001$], and the lowest number in low-quality patches [M = 5 ± 2, $t(40) = -10.196$, $p = 0.001$].

The same pattern of results was observed in the foraging gerbils. They also achieved more reward captures with increasing patch quality: $F(2,34) = 1052.868$. The reward yield in low-quality patches was 2 ± 0.3 rewards on average, in low-quality patches, 3 ± 0.3 in medium-, and 4 ± 0.3 in high-quality patches. The differences were statistically significant between high- and low- [$t(16) = -22.095$, $p < 0.001$], and high- and medium-quality patches [$t(16) = -10.240$, $p < 0.001$]. Point plots of averaged median rewards obtained as a function of patch quality (i.e., start reward probability) for both species are shown in Fig. 2a.

The fixed-time rule states that a forager would spend an equal amount of time in a patch regardless of the current intake rate or given patch quality. Thus, one would expect equal residence times across all three patch qualities. Trial durations did not differ between both human samples and were again averaged for the analysis, [fMRI: M = 41.992 ± 17.330; PC: M = 37.714 ± 17.617, $t(40) = 0.773$, $p = 0.221$]. Unlike the prediction of the fixed-time rule, humans' averaged median residence times were modulated by the reward probability, $F(2,82) = 66.955$, $p < 0.001$, with the longest average residence times of 45.594 ± 17.506 s in the high-quality patches, followed by 40.561 ± 17.453 s in medium-quality, and the shortest residence time [33.100 ± 19.100] in low-quality patches. Yet, post hoc tests showed that only residence times in high-quality patches were significantly higher than those in low-quality patches, $t(41) = -3.132$, $p = 0.006$; low vs. medium: Tukey's HSD $p = 0.152$, medium vs. high: $p = 0.421$).

Similarly, gerbils' residence times increased with patch quality, $F(2,34) = 38.761$, $p < 0.001$. The averaged median time that gerbils spent in the low-quality patches [13.675 ± 1.021 s] was on average 7.293 ± 1.342 (SE) s shorter than the averaged median time spent in high-quality patches [20.968 ± 4.138 s, $t(16) = -5.435$, Tukey's HSD $p < 0.001$] and the averaged median residence time in medium-quality patches [16.786 ± 3.324 s] was on average -4.182 ± 1.342 s shorter compared to high-quality patches, $t(16) = -3.117$, Tukey's HSD $p < 0.001$. Averaged residence times as a function of patch quality (i.e., start reward probability) for both species are shown in Fig. 2b.

Given the rapid depletion of rewards, subjects were encouraged to readily leave a current patch instead of spending too much time in it. Consistent with this, human residence times were negatively correlated with the total number of reward earnings (in €) they yielded in an entire foraging session [PC-lab: $r_{Pearson} = -0.717$, $p < 0.001$; fMRI-lab: $r_{Pearson} = -0.653$, $p = 0.002$] (see Fig. 2c, d). In other words, the more time subjects invested searching per patch, the less earnings they yielded throughout the entire experiment. No evidence for such a relationship was found in the gerbil data, [$r_{Pearson} = 0.280$, $p = 0.258$].

### Splitting groups by their median giving-up times

We observed a large variation in human participants' residence times [range = 66.936 s] and giving-up times [GUTs; range = 14.419]. This was in stark contrast to the very consistent gerbil data (range of residence time = 7.866 s; range of GUT = 3.580). In human participants, residence times strongly correlated with the GUTs, $r_{Pearson} = 0.729$, $p < 0.001$. Like human residence times, GUTs of both the PC- and the fMRI-lab samples correlated negatively with total earnings, $r_{Pearson} = -0.668$, $p < 0.001$,

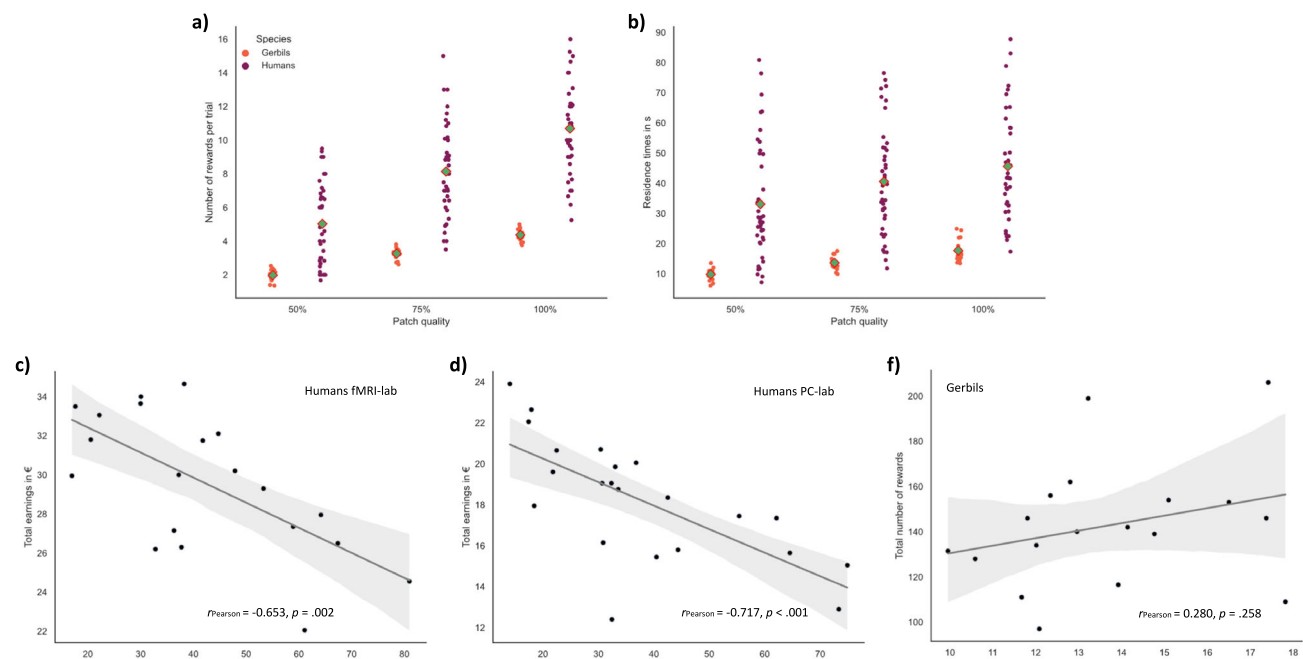

**Fig. 2 | Task performance measures. a** Dot plot shows the number of reward captures as a function of patch quality for humans (dark red) and gerbils (orange). Small dots indicate individual subjects' values and green diamonds are the sample mean. **b** Residence times for humans (dark red) and gerbils (orange) as a function of patch-quality. **c–e** Plots show the relationship between the total number of rewards obtained at the end of foraging sessions and residence times per patch. In both the PC-lab (**c**) and the fMRI-lab (**d**), humans' residence times were negatively correlated with total earnings. The association was reversed in gerbils (**e**) but was not statistically significant.

$r_{Pearson} = −0.676$, $p = 0.001$. No such correlation was found in gerbils, $r_{Pearson} = 0.350$, $p = 0.153$.

Thus, to better account for the heterogeneity in humans, we split the group by its median GUT (6.951) into a long- and a short-GUT group [long-GUT, $n = 20$: residence time: M $= 49.960 ± 14.729$ s, GUT M $= 10.414 ± 2.036$ s; short-GUT, $n = 20$: residence time M $= 30.415 ± 14.729$ s, GUT M $= 4.138 ± 1.842$ s]. Both residence times and GUTs for both subgroups are shown in Fig. 3a, b). Unsurprisingly, the long-GUT subjects had significantly longer residence times compared to short-GUT subjects, $t(42) = 4.412$, $p < 0.001$. GUTs were, on average, significantly longer in the fMRI subjects, [fMRI: M $= 8.510 ± 3.543$; PC: M $= 5.858 ± 3.378$; $t(40) = 2.414$, $p = 0.021$]. This means that more fMRI subjects entered the long-GUT group (15 out of 21 subjects), while more PC-lab participants were included in the short-GUT group (14 out of 21). Yet, importantly, within the short- and long-GUT groups, there were no differences in GUTs between fMRI and PC-lab participants, [long-GUT, $p = 0.145$, short-GUT, $p = 0.150$].

For comparison, we also split the group of gerbils in the same way into a long-GUT, [GUT M $= 5.690 ± 0.780$ s, residence time M $= 19.004 ± 1.127$ s], and a short-GUT group [GUT M $= 4.347 ± 0.390$, residence time M $= 15.282 ± 0.843$ s]. Also, in gerbils, residence times were significantly longer in the long-GUT group, $t(16) = 4.946$, $p < 0.001$, but in contrast to humans, the two groups of gerbils did not differ in the total reward captures, $p = 0.722$. Figure 3d shows the gerbils' residence times as well as GUTs as a function of patch-quality for both groups.

**Prolonged giving-up times indicate a bias for exploitation in humans**

An optimal GUT rule should account for differences in patch-quality with longer GUTs in better patches (McNair, 1982). In contrast to this prediction, already Fig. 3b) shows that the long-GUT humans invested the longest GUTs with an average of $11.033 ± 2.896$ in low-quality patches. Compared to this, GUTs decreased to $10.006 ± 2.422$ in medium-, and to $9.699 ± 2.127$ s in high-quality patches. Yet, a one-way ANOVA provided no evidence for true differences, Friedman F(1.905, 38.095) = 2.674,

$p = 0.084$. In the short-GUT group of humans, average GUTs were $4.250 ± 2.246$ s in low-, $3.959 ± 1.940$ s in medium-, and $3.769 ± 1.840$ s in high-quality patches. This time, we found a significant effect of patch-quality on GUTs, Friedman F(1.905, 38.095) = 3.333, $p = 0.049$. Post-hoc tests provided anecdotal evidence for different GUTs between high- and low-quality patches, $p = 0.054$

GUTs with comparable durations across patch types would still be in accordance with a simple GUT rule that does not account for differences in patch-quality. If participants applied such a rule, their GUTs should then consistently exceed previous time intervals between two consecutive reward captures (i.e., inter-target times, ITT). This is because participants using a fixed GUT to time the patch-leaving, would leave a patch before their ITTs exceed their GUTs because they only tolerate the fixed duration (i.e., the GUT threshold) without a new capture. Averaged ITTs and GUTs as a function of patch-quality are shown in Fig. 4a) for each long-GUT subject and in Fig. 4b) for short-GUT subjects. In the long-GUT group, all 21 subjects had averaged GUTs consistently longer than their average ICIs across all three patch types. This proportion dropped significantly in the group of short-GUT subjects, with only 6 participants displaying a GUT-ITT pattern in support of a simple GUT rule, proportions z-test: $z = −4.582$, $p < 0.001$. This pattern of results showed that long-GUT subjects' behavior was consistent with a fixed GUT rule in all participants of that group. Yet, their GUTs were prolonged to an extent where it affected the overall task performance negatively. In the humans' short-GUT group only less than half of the participants' data was in support of a GUT rule. Single gerbils' GUT–ITTs patterns showed that in the long-GUT animals, eight out of nine individuals had, on average, longer GUTs than ITTs, and in the short-GUT animals, only one individual had average ITTs exceeding its GUTs in medium-quality patches (see Fig. 4c, d). Thus, in both groups of gerbils, the data pattern was consistent with a simple GUT rule with no significant changes in proportions of animals behaving against the rule's prediction, $p = 0.303$.

Taken together, on average, all but one gerbil's GUT data was consistent with the predictions of a simple GUT rule. In humans, only the long-GUT group behaved in accordance with the simple GUT rule. Yet, unlike

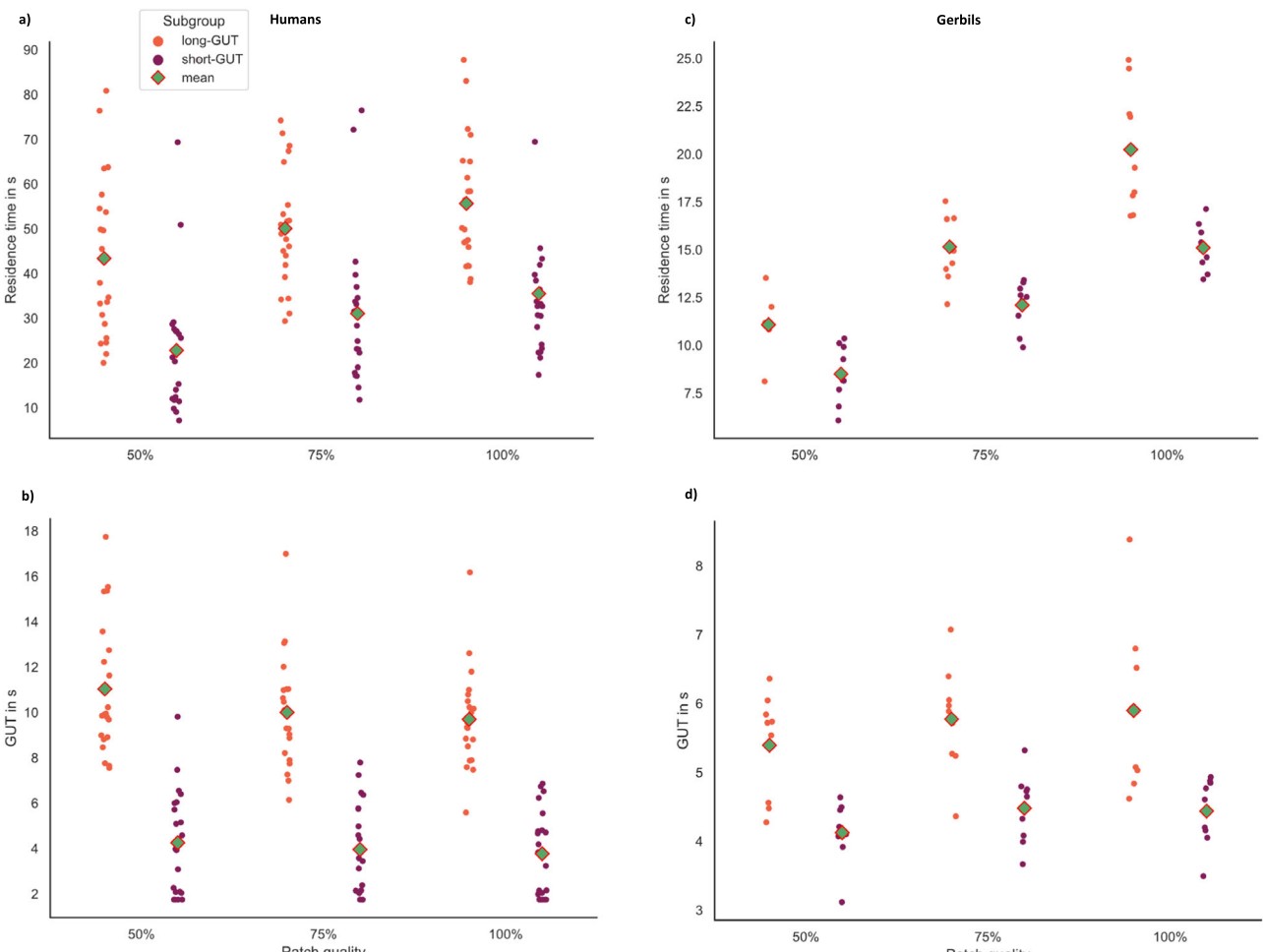

**Fig. 3 | Residence times and giving-up times as a function of patch-quality after group-splitting. a** Point plots show humans' residence times as a function of patch-quality. Small dots in orange indicate individual data points of the long-GUT humans and dark-red dots index individual data points of the short-GUT group.

Diamonds index the mean values. **b** GUTs of humans as a function of patch quality. **c** Gerbils' individual residence times after group splitting. **d** GUTs as a function of patch-quality are shown for the long- (orange) and short-GUT (dark-red) gerbils.

the animals, these human subjects seemed to choose suboptimal GUT durations. Given the task conditions, especially due to the quick depletion of reward, there was no benefit in prolonging residence times after the first few target encounters. Long-GUT humans who did this regardless showed significantly poorer task performance compared to short-GUT humans. This difference in performance did not exist between long- and short-GUT animals, likely due to the only marginal difference in GUTs between the two groups of animals.

**Increments in residence times following reward captures in both species**

Overall, the data provided good evidence for the GUT rule in both groups of gerbils as well as in the long-GUT human subjects. The short-GUT group of humans, however, showed data that were inconsistent with such a rule. Thus, we next examined the relationship between reward captures *within* a patch and residence times. Given the unpredictable changes in patch-quality, rewards encountered within a patch provide the only viable estimate of the underlying patch-quality. If foragers relied on this estimate, they would extend their residence times incrementally with each novel reward capture because each new reward encounter would suggest that the current patch is potentially of high quality. To test this, we again calculated within-subject regressions, but this time, we regressed the residence times on the number of reward captures[12,16]. This way, we obtained a slope and intercept for each participant, where the intercept represented the Initial time spent in

the current display without a reward detection, and the slope represented the increase in the residence time with each new reward capture.

Individual slopes in both subgroups of human participants were in all cases positive and, on average, significantly above zero, [short-GUT: mean slope = 3.372 ± 0.975, $t(20) = 15.457$, $p < 0.001$, long-GUT: mean slope = 3.748 ± 0.712, $t(20) = 23.528$, $p < 0.0001$] (Fig. 5a). This suggested that participants indeed extended their residence time in response to a new reward capture, consistent with the incremental patch-leaving rule. Within-subject slopes in the short-GUT group of humans, that had shown less evidence for a GUT rule, did not differ from the slopes of the long-GUT group, $t(40) = -1.390$, $p = 0.172$, indicating that the incremental effect of reward captures on the likelihood to stay in the current patch was comparable in both groups of humans. We conducted the same within-subject regressions also for the short- and long-GUT group of gerbils and obtained a similar pattern of results. The mean slope was 3.368 ± 0.690 in the short-GUT group, $t(8) = 13.795$, $p < 0.0001$, and 4.507 ± 0.674 in the long-GUT group, $t(8) = 18.900$, $p < 0.0001$. The average slopes of long-GUT gerbils were significantly larger than the average slope of the short-GUT gerbils, indicating a stronger incremental relationship in those animals with longer GUTs, $t(16) = 3.335$, $p = 0.004$ (Fig. 5b).

Although we anticipated that reward encounters affected residence times incrementally, we agree that one must be careful to not infer causation from correlation: regressing residence times on the number of rewards (residence times ~1 + reward encounter * $\beta$), we anticipated that the slopes

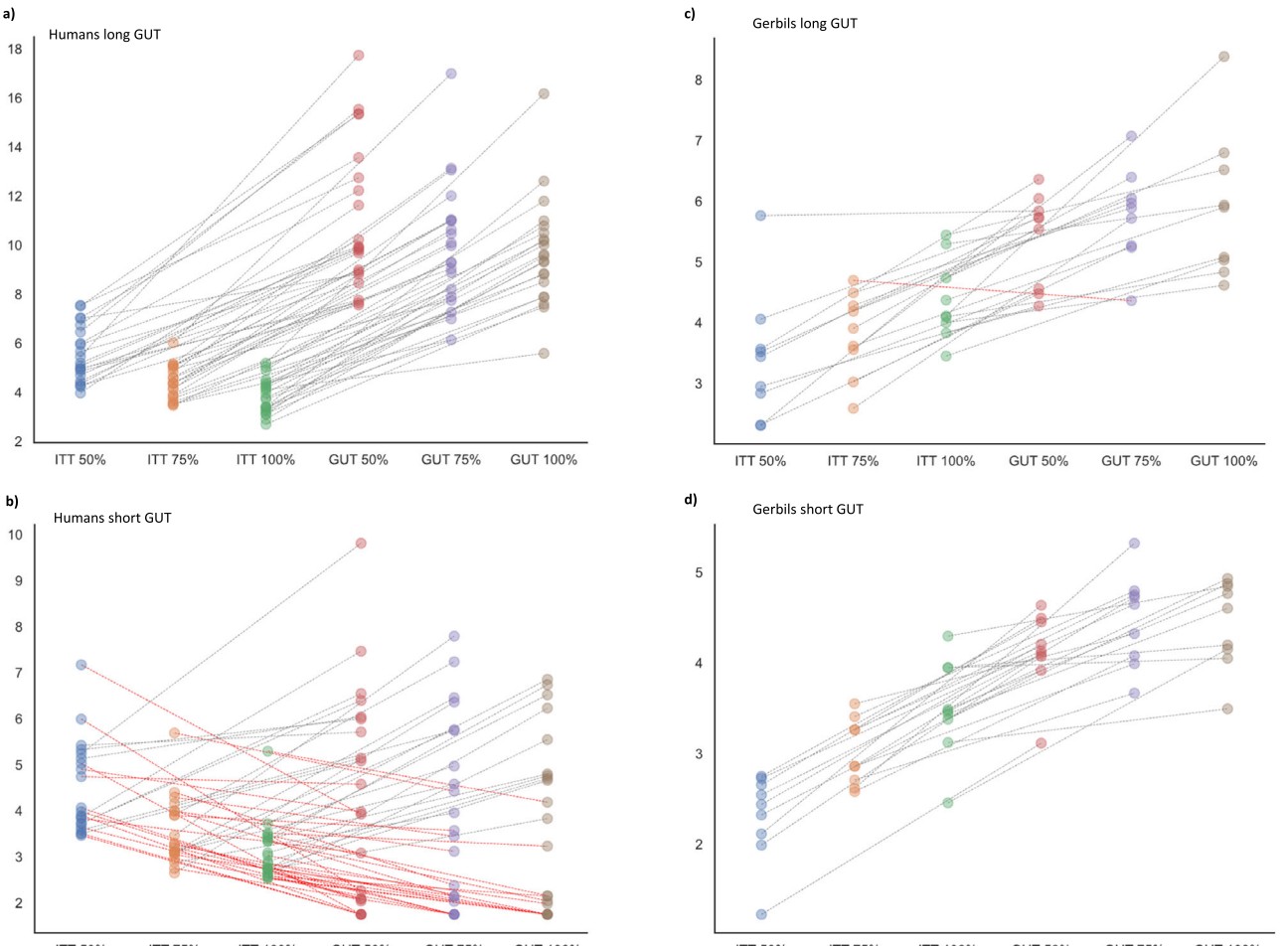

**Fig. 4 | Testing the GUT rule. a** Point plots show individual humans' GUTs and averaged time intervals between two consecutive reward captures (i.e., inter-target intervals, ITT) in seconds (s) plotted for each patch quality of the long-GUT group. Connecting lines indicate the values that belong to the same individual. In all subjects GUT durations consistently exceeded the average ITTs, in accordance with a GUT rule used for patch-leaving. **b** Corresponding GUT and ICI data shown for the

short-GUT group, red lines show a downward trend indicating 14 subjects who had lower GUTs than ITTs and, thus, a GUT-ITT pattern not conforming to the GUT-rule. **c** The point plot shows the same GUT-ITTs relation for the long-GUT gerbil; the single red line indicates the deviation from the GUT-rule in a single gerbil. **d** Point plots show short-GUT gerbils' GUT-ICI patterns. Here, all animals showed data consistent with the GUT rule.

would be positive, reflecting an incremental mechanism between the two variables. Yet, alternatively, it could be that longer residence times might simply increase the likelihood of finding more rewards, thereby creating a positive correlation between captures and time spent. In this scenario the time spent in a patch is random and not influenced by capture success. To test this 'random model' against our incremental model, we followed Hutchinso's approach[16] and took individuals' residence times as well as the initial reward probabilities and simulated the number of reward encounters on a trial-by-trial basis, assuming that reward captures are simply proportionally related to the residence times (number of reward captures = residence time * initial reward probability/100). In a second step, we conducted the same within-subject regressions, regressing residence times on the number of (simulated) reward captures on a trial-by-trial basis. Lastly, we compared the slopes obtained in the simulation (i.e., the increment in residence times by a reward encounter if the relationship was random) with the slopes observed in the real data. The results indicated significantly larger slopes in the real data, supporting the idea that participants indeed extended their residence time following each capture[16,17].

Simulations of reward encounters and the within-subject regressions were carried out 100 times for each subject. We then recombined the resulting slopes via bootstrapping on group-level using 100,000 bootstraps to get an estimate of the distribution of slopes on group-level that would be obtained assuming that the time spent in a patch is random and not

influenced by capture success. Because there was no indication of differences between long- and short-GUT humans, we carried out this control analysis jointly for both subgroups. The results are shown in Fig. 5a) and confirmed a smaller slope (mean slope = 2.138 ± 0.326, 95% bootstrap interval [2.137, 2.159]) obtained from within-subject regressions on simulated data in which the relationship between residence times and number of reward encounters was randomly proportional, $t(40) = -11.061$, $p < 0.001$.

The same analysis we also conducted for the gerbil data. Here, however, the additional analysis provided no further support for an incremental relationship between reward encounters and residence times (Fig. 5b), and the slope obtained from analyzing the simulated data (mean slope = 5.475 ± 0.252, 95% bootstrap interval [5.453, 5.480]) was even larger than the slopes based on the observed data, short-GUT gerbils $t(25) = 8.861$, $p = 0.999$, long-GUT gerbils $t(25) = 4.199$, $p = 0.999$.

Taken together, analyzing the association between reward encounters and residence times provided evidence for the incremental rule of patch-leaving (Fig. 1c) only in humans.

**Instantaneous and average collection rates—early gerbils, belated humans**
According to the marginal value theorem, optimal patch-leaving decisions are timed to the moment when the ICR approximates the MCR. As an estimate of the collection rate at which a reward capture $i$ occurred, we used

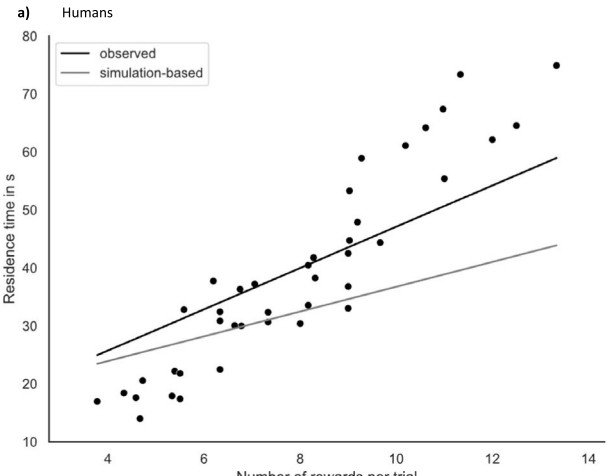

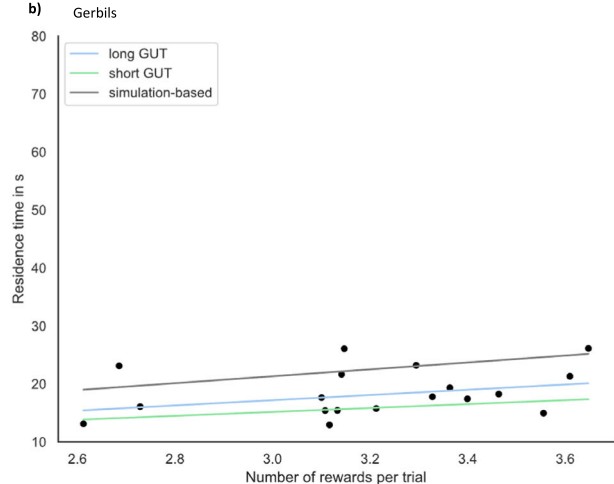

**Fig. 5 | Incremental relationship between reward captures and residence times.** **a** A dot plot on group level for all humans plotting individual averaged residence times as a function of a number of rewards per patch. Black dots index individual means. The gray line indicates the averaged empirical slope obtained from the within-subject regressions, regressing residence times on the number of rewards for both groups of humans. Subgroups were not distinguished further because the empirical slopes between short-GUT and long-GUT humans did not differ. The black line shows the averaged slope obtained in the same within-subject regression

analysis (residence time ~ reward) but performed on the *simulated* number of rewards. **b** The same dot plot for the gerbils, where each dot represents an animal's averaged residence time as a function of reward. Colored lines in blue (long-GUT) and green (short-GUT) indicate the averaged empirical slopes obtained in the within-subject regressions (residence time ~ reward). The black regression line indexes the average slope obtained from the within-subject regressions based on the simulated number of rewards.

the inverse of the time that had passed between the previous reward capture $i − 1$ and the reward capture $i$ (i.e., 1/intertarget times). To approximate the collection rate in the moment of patch-leaving, we used the inverse of the observed GUT, i.e., the time since the last reward capture and leaving the current patch[15]. The MCR we obtained by dividing the total number of reward captures by the total search time including travel times[4]. Given the difference in the number of total earnings between short- and long-GUT human subjects, we predicted that the latter group of participants had extended their residence times longer than what is considered optimal according to the MVT (i.e., estimated ICRs at the time of leaving should be significantly lower than the average collection rate).

Figure 6a shows the trajectory of the ICRs as a function of target captures for the long-GUT human subjects and b) for the short-GUT group. Testing the MVT prediction, a one-way repeated measure ANOVA with the type of time interval (ICRs for the three patch qualities as level 1–3, and the average collection rate as the level 4) yielded a significant main effect for the type of interval, $F(3, 60) = 102.502$, $p < 0.001$. In line with our prediction, post hoc contrasts showed that all three estimated collection rates at the time of patch-leaving were below the average collection rate of $0.165 \pm 0.02$ rewards/s, with $p < 0.001$ in all three patch types. Also, in the short-GUT humans we found a significant main effect of the type of time interval, Friedman $F(2.905, 58.095) = 3.457$, $p = 023$. However, post hoc tests revealed that in high-quality patches, estimated ICRs at the time of patch-leaving were still significantly above the average collection rate, $p = 0.015$, and no evidence for a difference was found in medium- and low-quality patches, with $p = 0.277$, and $p = 614$. Thus, at least in medium and low-quality patches, we found evidence for an optimal timing of the patch-leaving according to the MVT.

Repeating the same analysis for the gerbil data, the one-way repeated measure ANOVA with the type of time interval as the single factor yielded a significant main effect in both groups of gerbils, [long-GUT: Friedman $F(2.778, 22.222) = 8.701$, $p < 0.001$; short-GUT: $F(2.778, 22.222 = 19.0, p < 0.001$]. Post hoc contrasts for the long-GUT gerbils showed that in low-quality patches, the estimated collection rates at the time of leaving were still significantly higher than the MCR, $p = 0.001$. No evidence for such differences was found for medium- and high-quality patches, $p = 0.185, p = 0.670$. In short-GUT gerbils, ICRs in low- and high-quality patches were still significantly above the MCR, $p = 0.001, p = 0.010$, in medium-quality

patches the evidence was anecdotal, $p = 0.052$. Figure 6c, d shows the trajectory of the ICRs as a function of target captures for the long-GUT and the short-GUT group of gerbils, respectively.

## Formal model testing of cues used to inform patch-leaving decisions

Lastly, we used Cox regressions to test different predictors that potentially increased or decreased the likelihood of staying in the current patch[16]. For this purpose, we used cox-regressions that model the impact of different factors on the probability of leaving the current patch (i.e., the hazard ratio of the patch leaving). The Cox proportional hazard model is a regression model typically used in epidemiology to find out the relationship between the survival time of a patient and one or more predictor variables[18]. The model has also become widely used in the foraging literature[7,16] to model the residence times using the following hazard function:

$$h(t) = h_0(t) * \exp(b_1 x_1 + b_2 x_2 + \ldots + b_n x_n)$$

where $t$ is the residence time, and $h(t)$ the hazard function of the residence time, $b$ indicates the impact of the predictor $x$ on the probability to reside in the current patch. The resulting value $\exp(b_i)$ is called the hazard ratio for the predictor $i$. It refers to the relative 'risk' of leaving the current patch for different levels of the $i$th predictor. In other words, it quantifies the change in risk of leaving the current patch associated with a unit change in the $i$th predictor. A hazard ratio greater than 1 indicates an increased risk of leaving the patch, while a hazard ratio less than 1 indicates a decreased risk. A hazard ratio of 1 indicates no change in risk.

The within-subject regressions with the number of rewards captured as the predictor and the residence times as the outcome had already indicated an incremental relationship between the two variables, consistent with an incremental mechanism based on reward encounters driving patch-leaving decisions in both species. Given this finding, we entered the number of reward captures as the first predictor of the model and expected a hazard ratio below 1, i.e., a protective effect of the number of rewards decreasing the risk of patch-leaving. Since all humans and gerbils had positive slopes in the within-subject regressions, we expected this protective effect to be significant in both short- and long-GUT humans and animals. In addition, we used the averaged inter-target times (ITT) between the last and the second

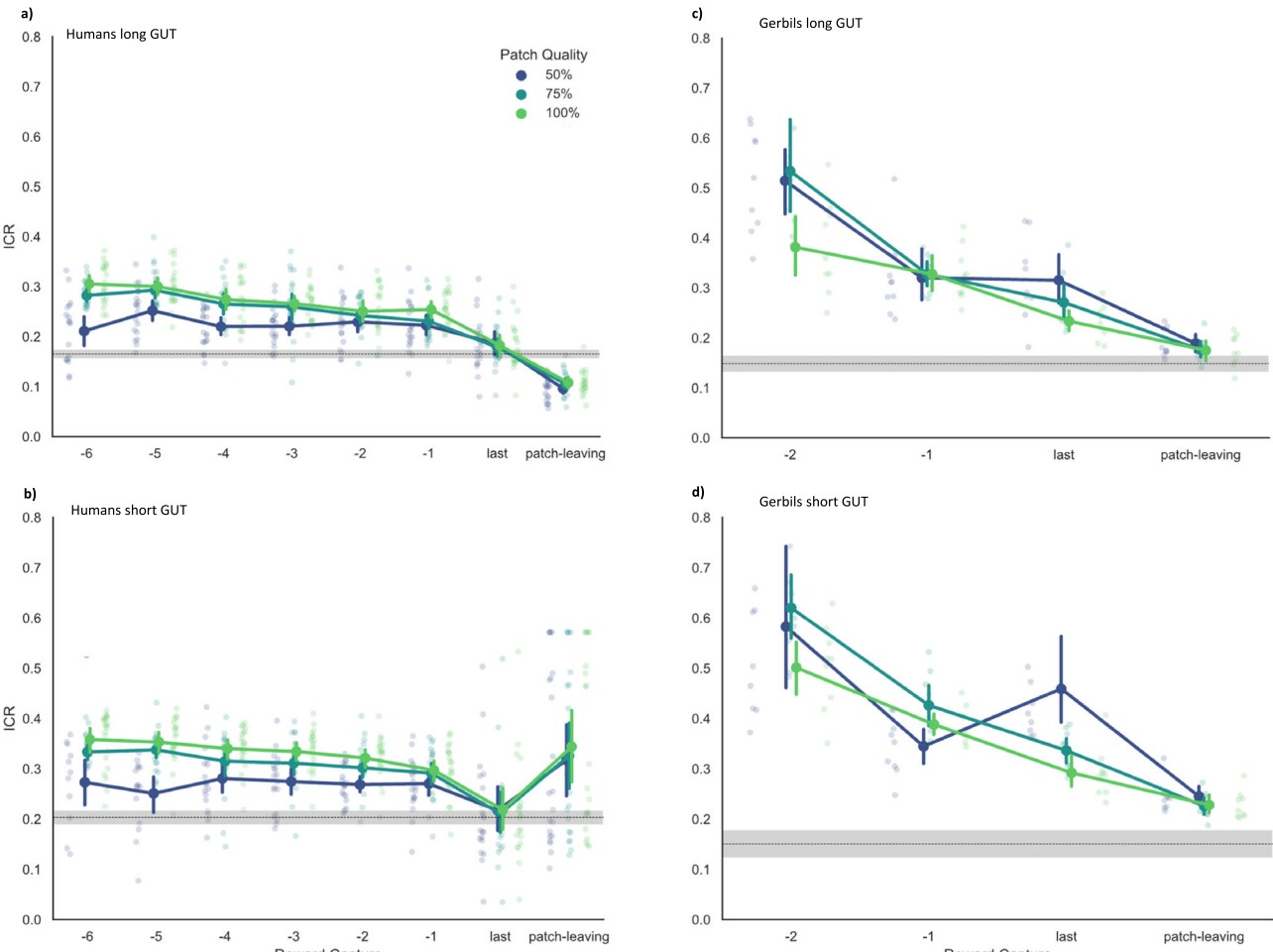

**Fig. 6 | Collection rates.** Point plots show the ICRs for the seven last reward captures averaged for the long-GUT (**a**) and short-GUT humans (**b**). Error bars equal ±standard error. The gray dashed line marks the averaged overall collection rate given by the number of total rewards divided by the total search time in seconds (the shaded area indicates the 95% confidence interval). The estimated capture rate at the time of leaving was defined as the inverse of the participant's GUT with short GUTs leading to higher estimated ICRs at the time of leaving compared to long GUTs. Point plots show ICRs for the long-GUT (**c**)) and short-GUT (**d**)) gerbils of the last three reward captures. Note that only the last three reward captures are plotted here because this was the average number of rewards obtained. In contrast to the MVT prediction, ICRs in gerbils were well above the MCR (red dashed line) at

the time of leaving (i.e., the ICR at which the last reward was captured). This estimate of collection rate in the moment of patch-leaving (=1/GUT) is independent of actual reward encounters, relying solely on the participants' propensity to continue foraging unsuccessfully. A lower GUT implies a higher estimated collection rate at patch-leaving, potentially even leading to an increase of the rate compared to the collection rate at which the preceding rewards had been captured, as seen in Fig. 6b, where the estimated CR at patch-leaving was higher than the CR of the last and second last reward (i.e., '−1' on the x-axis). It is essential to note that this estimate does not originate from an actual reward encounter and, thus, does not violate the assumption of a depleting patch.

before-last reward captures and between the second-last and the third-last reward captures. This value could provide subjects with a good estimate of the recent collection rate. Increases in value should increase the risk of leaving, at least in the short-GUT group of humans that had shown collection rate data most closely in accordance with the MVT[16]. Thus, we expected a hazard ratio significantly larger than 1 in the short- but not in the long-GUT group of humans, indicating that only the former group of subjects used their recent reward capture rates as a cue for patch leaving.

In line with our prediction, the Cox regression for the short-GUT humans revealed that, while controlling for the number of rewards, the hazard ratio for the averaged ITT was 5.01, 95% CI [1.30, 19.42], indicating a substantial increase in the 'risk' of patch-leaving, Wald $\chi2(1) = 2.33$, $p = 0.02$. In other words, if the average ITTs of the last two target captures increased by 1 s, subjects were 5 times more likely to leave the current patch compared to no increase in the averaged ITT. Thus, consistent with the MVT, the Cox model confirmed that the short-GUT humans were sensitive to declines in their current collection rates and timed their patch-leaving accordingly. Again, in line with our prediction, the number of reward

captures conversely appeared to be a protective factor regarding the 'risk' of patch-leaving as they decreased the risk of leaving by 80% with each new reward capture, (hazard ratio = 0.20, 95% CI [0.09, 0.44], Wald $\chi2(1) = -3.95$, $p < 0.005$), consistent with the incremental rule.

Next, we computed the same Cox regression model also for the long-GUT group. Again, we could confirm the results of within-subject regressions that had indicated an incremental relationship between reward captures and residence times in that the number of reward captures had a protective effect on the 'risk' of patch-leaving, decreasing the risk by 73%, hazard ratio = 0.27, 95% CI [0.13, 0.55], Wald $\chi2(1) = -3.59$, $p < .005$. Intriguingly, in the long-GUT group, the average ITT had no effect on the 'risk' of patch-leaving, hazard ratio = 3.89 95% CI [0.38, 39.63], Wald $\chi2(1) = 1.15$, $p = .25$. Thus, in the long-GUT group, subjects' estimates of the recent collection rate did not impact their patch-leaving. These findings are consistent with results from the collection rate data showing that long-GUT humans' ICRs were not in agreement with the MVT.

The data obtained from gerbils with long- and short-GUT were much more consistent between the two groups. We did not find any significant

differences in the effect of the number of rewards and averaged ITTs on the risk of gerbils' patch-leaving between the two groups. Hence, the Cox regression was carried out for the entire group of gerbils (pooled long- and short-GUT animals together). The results matched those of the humans in the short-GUT group: if the averaged ITT increased by one second, gerbils were almost 3.5 times more likely to leave the current patch, hazard ratio = 3.46, 95% CI [1.03, 11.69], Wald $\chi 2(1) = 2.00$, $p = 0.05$, while a new reward capture decreased the 'risk' of leaving by 94%, hazard ratio = 0.06, 95% CI [0, 0.90], Wald $\chi 2(1) = -2.04$, $p = 0.02$, confirming that gerbils relied on the reward encounters they experienced (incremental rule) but also on their current collection rates (MVT) in order to make patch-leaving decisions.

Restricted access to food facilitated that the animals started each daily session with appetite and motivation. However, it is important to consider that this motivation might diminish with satiation. Such a decrease in motivation could lead to reduced task commitment, potentially resulting in a higher occurrence of task-unrelated behaviors. This, in turn, could notably affect behavioral measures such as the average collection rate. To explore this hypothesis, we performed a split-half analysis within each session, comparing the behavioral parameters of gerbils between the two session segments and across the three levels of patch quality. These results indeed revealed behavioral changes consistent with a decrease in task motivation. While the number of nose pokes remained constant throughout an experimental session, increases in the inter-poke intervals, residence times, and travel times suggested an increase in the frequency or duration of task-unrelated behaviors such as grooming. Details of this analysis are reported in the supplementary material, S1, and Supplementary Fig. 1. However, crucial parameters describing the animals' patch-leaving behavior suggested that animals continued to optimize the timing of their patch departures despite the fading task motivation. Although the average collection rate declined as a function of session split and GUTs increased in the second compared to the first half of a session, the difference between ICRs and MCRs was, on average, significantly smaller in the second half. Importantly, these results do not challenge the conclusion that we made based on the results reported previously. Details of this analysis are reported in the Supplementary Material, S1.1, and Supplementary Fig. 2.

## Discussion

This study had the goal of elaborating a probabilistic foraging paradigm for inter-species comparisons. To this end, we designed two separate foraging tasks tailored to suit each species. Although the tasks differed in their specific details, the underlying reward structure was standardized in both tasks. This allowed us to investigate both the similarities and dissimilarities in patch-leaving behavior between animals and humans during foraging.

Timing patch departures based on a fixed number of reward captures or based on a fixed amount of time only works well if the forager roams an environment that offers patches that do not differ greatly in quality. However, both the fixed N- as well as the fixed-T rule are not optimal if patches within an environment differ greatly in quality[13,19]. Thus, given the randomly changing reward probabilities in our paradigms, the use of these rules would have been disadvantageous for our species. The findings that both residence times and reward captures increase with increasing patch quality, confirmed this assumption. However, humans showed high variability in individual residence times and GUTs. We, therefore, divided the sample of humans based on the median GUTs into long- and short-GUT individuals to evaluate whether different rules apply to these two subgroups of human participants. For better comparison we did the same also for the group of gerbils. Only human subjects showed evidence for the incremental rule of patch-leaving (see Fig. 1c), whereas both species, except for the long-GUT humans, timed their patch-leaving optimally according to the MVT.

Being confronted with unknown and randomly varying initial reward probabilities, single reward encounters provided a first means for the gerbils and humans to learn about the quality of the current patch, where each new reward encounter could be perceived as an indication that the current patch may be of good quality, and, thus worthwhile to spend more time in it.

Consistent with this notion, the within-subject regressions of residence times on the number of reward captures, showed a positive slope in all individuals of both species, suggesting that residence times were extended incrementally following a new reward capture[13]. However, longer residence times could have simply increased the likelihood of finding more rewards, resulting in a positive correlation between captures and time spent. In this case, the time spent in a patch is random and not causally driven by capture success. An additional simulation analysis supported a causal relationship (i.e., the incremental rule) only in humans but not in gerbils. The slopes from data simulated with the assumption of a randomly proportional relationship between reward encounters and residence times were less steep than those slopes obtained from within-subject regressions on the actual data, suggesting that the human participants did indeed increase time in a patch with each reward capture, confirming previous findings[12]. In gerbils, an analogous simulation analysis did not confirm this notion, inconsistent with the results in mice by Lottem and colleagues[7]. The authors analyzed mice's nose-pokes and fitted their data with a proportional hazard model. The estimated hazard rate reflected the probability of leaving a current patch as a function of nose-pokes that started at its minimum at the beginning of a trial and would increase with each unrewarded nose poke. Each rewarded nose poke, however, decreased the hazard of leaving, prolonging residence times, like our results in the Cox regression, where new reward encounters decreased the 'risk' of leaving. While the mice collection rates at the time of leaving were in keeping with the MVT, the incremental model showed a significantly better fit compared to an MVT-based model fitting in Lottem's study[7]. Thus, using the same foraging paradigm in gerbils, it will be important to test if those findings can be replicated when analyzing nose-pokes instead of residence times and reward encounters. At this point, only our human data confirms the incremental rule driving patch-leaving decisions.

The long-GUT humans also showed GUT data most consistent with a simple GUT rule: their GUTs appeared to be consistent across patch-types, and always exceeded the previous ICIs (see Fig. 4a). These subjects had adopted a rather detrimental GUT rule with excessively long GUTs resulting in overharvesting (i.e., exploiting a patch longer than what is considered optimal according to the MVT). The disadvantageous timing of their patch-leaving became evident in the significantly poorer overall performance measured in terms of total reward earnings compared to short-GUT humans. In the short-GUT group of humans, in contrast, only a minority of subjects demonstrated foraging behavior with GUT rule-conforming GUT-ICI patterns. Hence, the strongest evidence for a fixed GUT rule in humans was found in subjects who had the tendency to overharvest. Unlike the humans, both subgroups of gerbils had consistent GUTs across patch-qualities, and GUTs exceeded the ICIs in all but one animal, confirming a simple GUT rule. In accordance, also previous studies in other foraging animals facing patches of unpredictably varying quality reported patch-leaving behavior that agreed with a GUT-rule[20,21].

A mundane explanation for the overharvesting occurring exclusively in humans might be that these individuals possibly failed to understand the task. However, neither subjects' responses to post-briefing questions, nor their prior training performance supported this hypothesis. Moreover, prior to the main experiment, all subjects were told that an exhaustive search strategy (i.e., trying to find all existing rewards per display) would be detrimental to the overall task performance. Studies in elderly foraging humans reported age-related increases in GUTs as an indication of an increased behavioral tendency to exploit[22]. This suggests that exploration and exploitation as opposing behavioral tendencies together form a continuum and that individuals may differ in their position along this continuum due to age differences and other factors. Consistent with this, a recent study in patients with opioid-use disorder showed that the interindividual variability in overharvesting (in both users and controls) was related to a poorer neuromelanin signal, and indirectly catecholaminergic function (i.e., dopamine), of the ventra tecmental area[23]. Neurotypical subjects may already differ in their tendency to either explore or exploit based on genetic variations in those genes that control, e.g., the formation of the

catecholaminergic system. Still, the extent to which long-GUT subjects tended to overharvest is striking, given the disadvantage that arose from this strategy and more research is needed to further examine potential reasons that could explain these interindividual differences we observed.

The MVT theory posits that foragers decide when to leave a patch based on diminishing returns. Specifically, they should leave the current patch when the ICR—representing the current rate of rewards obtained—falls below the MCR of the entire environment. This timing ensures that the energy spent on acquiring additional rewards does not surpass the benefits gained and is thus considered optimal. Notably, the overharvesting long-GUT humans displayed a bias toward exploitation, which resulted in estimated ICRs at the time of patch-leaving significantly below the average collection rate. In contrast to this, the short-GUT humans, who tended to leave patches earlier, aligned with MVT predictions with estimated ICRs comparable to the average collection rate. Intriguingly, this latter group of subjects included participants of whom less than half exhibited ITT patterns conforming to the GUT-rule. Thus, our findings suggest that while the most compelling evidence for a maladaptive GUT rule was found in humans with above-median GUTs, those who left patches earlier in time showed collection rate patterns more consistent with MVT principles than with the GUT rule. This implies that the two groups may have employed different cues to determine their patch departures. In the case of the GUT rule, a subjective temporal threshold for unsuccessful searching is employed, whereas subjects adhering to MVT principles may have used recent ICRs as an estimate of their present intake rate and a cue for timing the patch-leaving. This hypothesis is supported by our Cox regression results in short-GUT subjects, where the average of the last two inter-target intervals – an effective estimate of the current ICR-—emerged as a robust positive predictor for patch-leaving. Increasing this interval (reflecting a decrease in the estimated ICR) heightened the 'risk' of leaving the patch. Conversely, within the long-GUT group, the same predictor (current ICR) yielded no statistical significance.

A recent foraging study used a visual search paradigm similar to our human task and reported that human participants foraged longer in a given patch than predicted by the MVT[3]. The participants performed either a conjunction search task (i.e., targets were defined by a combination of two features as in our experiment) or a feature search task (i.e., targets were defined by a single feature), and the results indicated similar foraging behavior for both search types. Yet, in contrast to the conjunction search task used in the present study, the volunteers were allowed to switch between target types within a given patch. During these switches the ICR would drop well below the MCR, but the subjects stayed in the same display, inconsistent with the MVT prediction. Additional investigations employing visual search paradigms, such as virtual berry-picking experiments, have also revealed deviations from the MVT's predictions under specific circumstances. The deviations are notably pronounced when patch quality exhibits large variability, and when visual information becomes impaired to the extent that foragers are unable to discern whether a target item offers a reward[4]. This evidence suggests that by rendering foraging tasks more complex, e.g., by allowing changes between search types[3] or introducing a high degree of reward variability[4], patch-leaving behavior appears to be no longer in accordance with the MVT. In humans, we showed that inter-individual differences in displaying a behavioral bias to exploit affect whether patch-leaving conforms to the MVT or not. A stronger exploitation bias leads to patch-leaving decisions that are less in keeping with the MVT in humans.

Testing the MVT in gerbils and animals in the long-GUT groups showed collection rates more aligned with MVT predictions. Only in low-quality patches these animals left the current patch when ICRs were still significantly above the MCR. Conversely, short-GUT gerbils demonstrated a higher propensity for early patch-leaving, with ICRs at the time of departure still significantly above the average rate in two out of three reward conditions. Thus, while gerbils' ICRs and MCRs suggest that they tended to leave patches slightly earlier than predicted by the MVT, the Cox regression analysis indicated that decreases in the ICR increased the likelihood of patch-leaving, consistent with MVT principles. Previous studies have

provided mixed evidence regarding the MVT, with some supporting it[7] and others not[24]. Our data suggest that although the gerbils' ICRs were not perfectly aligned with the environment's MCR at the time of patch-leaving, they were still sensitive to changes in their ICRs, as conformed by formal model testing, aligning with the predictions of the MVT.

A potential confound in our study could have arisen from differences in how reward probability decreased in the two paradigms. In the gerbil task, the reward was depleted with each executed nose-poke, whereas in the human task, depletion occurred following a new reward capture but not merely a target fixation, which would have been analogous to a nose-poke. Consequently, the reward decay following nose-pokes led to a faster depletion of the remaining reward probability. This might explain why the gerbils left patches more readily, resulting in very short residence times and a lower number of reward captures per patch compared to humans. Importantly, however, the unpredictable variability of reward distribution across patches was identical in both experiments, ensuring that both species faced similar environmental challenges with aggregated reward distributions. Therefore, despite the differences in reward decay mechanisms, we are confident that the two tasks were sufficiently similar to allow for a comparative study of patch-leaving decisions in humans and rodents. Nonetheless, future investigations should aim to standardize the reward decay process across both tasks to eliminate this potential confound and further align the experimental conditions.

Lastly, it needs to be noted that the high-GUT vs. low-GUT groups were introduced for data analytic reasons, i.e., to compare two extreme groups. Importantly, these two groups were no natural subpopulations resulting from the observed variance structure. The group splitting was rather a means to deal with the large variance in humans' residence as well as giving-up times data. Human subjects received only a minimum of training to familiarize them with the task. Thus, when starting the experiment, their performance may have not yet stabilized, introducing more within-subject noise compared to the intensively trained and, thus, more consistently behaving animals. Additionally, inter-individual differences in the propensity to explore versus exploit could have been driving the large dispersion in participants' GUTs and residence times. Even though the optimal strategy was to leave patches rather early, subjects' disposition to either persist or to leave more readily certainly impacted their behavior and introduced the large dispersion in both parameters. In support of this notion, a large study in humans recently confirmed that ADHD symptom-like behavior assessed by self-report predicted the residence times during a patchy virtual foraging task[25]. The higher a subject scored on an ADHD self-report screening assessment, the shorter their residence times were. This finding not only offers an interesting perspective on the evolutionary benefits that ADHD traits may offer, but it also provides an explanation of why inter-individual differences in strategies during a virtual foraging task may be large, especially if participants had not undergone extensive training before.

To better understand the neural coding mechanisms underlying patch-leaving decisions, it will be crucial to use fMRI in humans and recordings of local field potentials in gerbils. A central structure enabling attentional exploration in humans is the anterior prefrontal cortex, mainly consisting of Brodmann area 10 (aPFC)[26–29]. Lesions of the aPFC, both in human and non-human primates[30,31] prevent the exploratory allocation of attentional resources to novel aspects of the environment, thereby preventing optimal adaptation to the environment. Although aPFC has a causal role for exploratory attention shifts in primates, rodents lacking a distinct aPFC[32], undeniably show exploratory behavior[33]. This leads to the question of how the rodent brain supports exploratory and rule-based (conditioned) attentional resource shifts. Rodent experiments are therefore not only of interest from a phylogenetic perspective but also open opportunities for mesoscale investigations. Moreover, as the behavioral data reported here suggest that humans display strong interindividual differences in their tendency to exploit a current patch, future studies should target the question of whether such behavioral differences also translate to physiological differences, such as differential activations of the aFPC and other areas of the frontoparietal attention network.

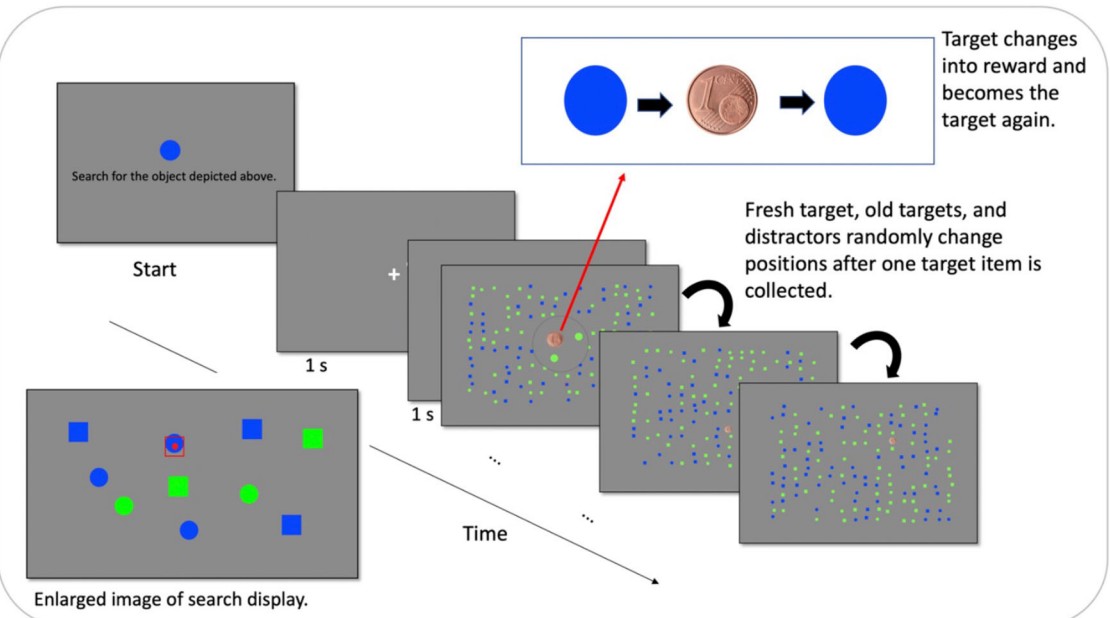

**Fig. 7 | Human visual search task.** Diagonal sequence represents a trial. At the beginning of a session, the target object (here a blue circle) was introduced. The beginning of a trial was cued by a central fixation (1 s) followed by a blank (1 s). Next, the search display (i.e., patch) appeared. By navigating the mouse cursor to a target (red square with red dot at center in lower left), participants realized a reward capture. Upon such a capture, the collected target turned into a reward for 500 ms (euro-cent image) and then changed back to its previous appearance. An already collected target would not turn into a reward again if fixated again. With each reward capture, all items changed positions randomly. At any time, participants were able to switch to a new display by button-press.

The present findings indicate that both species adapt to changing reward probabilities by avoiding disadvantageous rules of thumb for patch-leaving, using novel reward encounters to extend residence times. However, a subset of humans exhibited overharvesting, leading to prolonged GUTs and reduced earnings, a maladaptation absent in the rodents. Optimally timed patch-leaving was observed in non-overharvesting humans and gerbils, aligning with the MVT. These results underscore the value of inter-species comparisons but also highlight the variability in human foraging decisions, suggesting individual differences play a role in adherence to ecological models. Altogether, this study paves the way for future research comparing the neural substrates involved in exploratory decision-making between species, contributing to our understanding of the phylogenesis of cognitive control structures in the brain.

## Methods

### Statement of compliance with ethical regulations
All methods were performed in accordance with the German animal welfare law, the local ethics committee of the Otto-von-Guericke University, and the State of Saxony-Anhalt.

### Human study
**Participant.** In total, 52 (17 male) native German speakers participated in the experiment. Thirty-two of the participants were tested in the PC-laboratory (lab), while 20 subjects were tested in a follow-up neuroimaging experiment using functional magnetic resonance imaging (fMRI-lab). For this report, we used the behavioral data of the fMRI subjects to increase the original sample size. All volunteers were between 19 and 37 years old (M = 24.25 years), right-handed by self-report except for two participants, and had normal or corrected-to-normal vision. They provided written consent consistent with the protocols approved by the local ethics committee of Otto-von-Guericke University prior to the experiments and were monetarily reimbursed based on the earnings they made performing the foraging task. We excluded eight PC-lab participants from the data analysis because they performed less than six trials in at least one of the reward conditions. Another two PC-lab subjects who yielded an overall number of trials below the 1st or above the 99th percentile of the group distribution were also excluded. Thus, the final sample size was 42.

### Visual search paradigm
**Set-up and Stimuli.** We used the Python toolbox "PsychoPy"[34] to control the stimulus display and responses. The stimuli were presented on a 24″ Samsung monitor (1920:1080 resolution, 60 Hz refresh rate). All participants were positioned 50 cm away from the screen. Stimuli consisted of geometrical forms. These were either squares or circles that appeared in either blue or green color. All stimuli subtended 0.59° visual angle. Their spatial locations were randomly assigned on a spatial grid spanning a rectangle field of 12.9° * 14° visual angle. One stimulus type (e.g., blue circles; Fig. 7) was assigned as a target, while the three remaining stimulus classes would serve as distractors. As a reward indication, we used an image of a Euro-Cent symbol that subtended 0.70° visual angle.

**The experimental task.** To study the exploration-exploitation dilemma in human subjects, we designed a probabilistic foraging task. Participants were asked to search and collect target items among distractors in a visual feature conjunction search task. Stimuli consisted of simple square- and circle-shaped objects randomly located in the search display. They used the mouse to navigate through the display. Target items were defined by a specific conjunction of shape (i.e., circle) and color (i.e., blue), and equal numbers of distractors would differ either in shape (blue squares), in color (i.e., green circles), or in both feature dimensions (i.e., green squares). The total search time was restricted, and a countdown timer was constantly visible to the participants at the left bottom corner of the display. To obtain a reward, participants had to navigate the mouse pointer to a target. Once a target had been fixated for 300 ms, the target turned into a reward indicator (i.e., Euro-Cent) and then returned to its previous appearance. This served as the feedback that the target had been "foraged", and a reward was received. The participants were then able to continue the search for the next target in the display. At the display's left

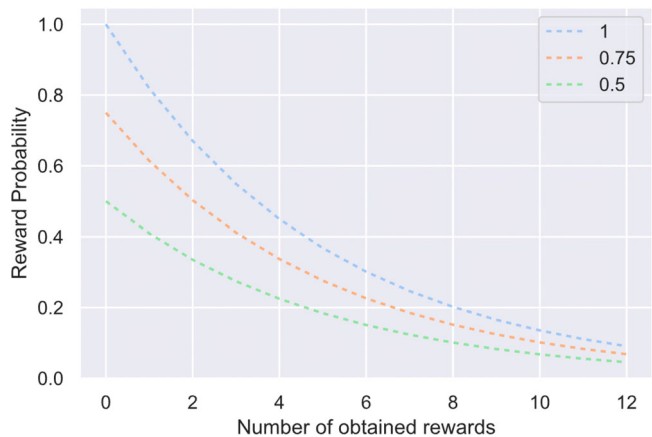

**Fig. 8 | Reward probability protocol.** Graph shows the decaying reward probability as a function of obtained rewards (i.e., patch) for the three reward conditions (100%, 75%, 50% initial reward probabilities – include label into legend). The exponential decay function was adopted from Lottem et al. [7].

bottom corner, the participants were able to constantly track the total number of rewards they had already earned. At the display's right bottom corner, they could keep track of the remaining time.

Once a collected target returned to its previous appearance, it was turned inactive so that a second fixation of the same target would not result in a further reward capture. Moreover, with each collected target item, additional targets, randomly located in the display, were also deactivated. In this way, after fixation on a new target, the remaining reward probability in a given display decreased exponentially, mimicking a quickly depleting food source. Furthermore, the whole spatial configuration of target and distractor locations changed after each collected target, and all targets and distractors consequently appeared at new locations. This manipulation made the search increasingly difficult and quickly inefficient as it was impossible to remember target locations that had been already visited and, thus, would not promise a new reward following fixation. To compensate for this, participants could choose to end the search in a display and to proceed to the next display at any time. The countdown did not pause when volunteers were directed to the next display, and each switch to a new display consumed time (3.5 s in the PC-lab, ~5.75 s in the fMRI-lab; see Task design) analogous to patch-leaving costs in ecological foraging or the movement from one foraging spout to the other in the gerbil experiment. After volunteers pressed the spacebar to continue to a new display, a central fixation cross appeared for one second, followed by a blank screen for another second before the new display appeared. With the appearance of a new display, the fixation cross (mouse point) was relocated to the display center. A depiction of a trial sequence is shown in Fig. 7.

**Task design.** We aimed to determine the optimal task conditions to study patch-leaving behavior in humans. This required a task in which volunteers actively decide to leave an exploited patch and switch to a new patch. Thus, residence times (i.e., trial durations) varied and were given by the time between entering a new display and switching to the next by button press. Travel times were given by the time between the onset of the button press and the occurrence of a new display. This exploratory foraging behavior is facilitated if the overall number of search targets in each search display is high and if the travel costs moving from one display to another is relatively short[4,35,36]. Therefore, we chose a total number of 40 targets and a relatively short travel time of 3.5 s. Due to additional intermediate data storage during the travel in addition to generating the upcoming display, travel times in the fMRI experiment were, on average, 5.75 ± 0.1 s.

To make reward encounters dependent on the foregone foraging success we chose an exponential decay function for the depletion of reward

following each reward capture[7]. That is, the number of available rewards drastically decreased within a relatively short period of time, resulting in an inefficient search. Due to the time constraint, participants decided to switch to a new display to improve search efficacy. The following decay function was adopted from Lottem et al.[7] (see Fig. 8):

$$P(o_n = 1|t_i) = A_i e^{(-(n-1))/5} \qquad (1)$$

here $t_i$ is the $i$th trial type, i.e., low-, medium-, and high-quality trials. The different trial types had three exponential scaling factors A1 = 0.5, A2 = 0.75, A3 = 1. $N$ indicates the number of already achieved reward captures (previous target fixations that resulted in an earning) within a trial. $O_n$ is the positive outcome of the $n$th target fixation (1 for reward). We additionally varied the initial reward probabilities by applying three conditions from high, middle to low probabilities to be able to test whether subjects adapted their patch-leaving strategies according to probabilistic changes in the environment. In the high reward condition (i.e., high-quality patch, blue function in Fig. 8), all 40 target items (100%) were active and would turn into a reward following a first fixation. In the medium reward condition, only 75% of all targets (i.e., medium quality patch, orange function in Fig. 8) were active and were associated with reward following a fixation. Only 50% of all targets were initially active in the low reward condition (i.e., low-quality patch, green function in Fig. 8).

The participants started the experimental session with two training trials in which they searched for target items in two consecutive displays without the time constraint, and no reward was registered. Once volunteers terminated the search in the second display, they were informed that the main experiment would start next and that they were given a total search time of 30 minutes (PC-lab), or 6 ×10 minutes (fMRI-lab), respectively. The fMRI session took place on a single day, and participants were able to take short breaks between the runs while remaining in the scanner.

### Animal study
Animal experiments were performed with 18 adult male Mongolian gerbils (Meriones unguiculatus, in-house bred). The age of the animals during these experiments varied between three to four months. All experiments were performed in accordance with the German animal welfare law (NTP-ID: 00041189-1-X).

**Food restriction.** The animals had free access to water but were food-restricted starting three days before the beginning of the foraging task. Before food restriction was started, the animals' body weight was measured over three days to obtain an average baseline body weight (BBW). The BBW of the animals was 70-80 g before starting the foraging task. To keep the animals' body weights above the critical level (85% BBW) during the foraging task period, food was supplemented inside the cage at least 2 hours after the end of the foraging task. The total daily food intake, including the amount of food retained during the training session of an animal, was between 3-7 g based on the performance of the animals in the foraging task.

**Foraging setup and stimuli.** Foraging tasks were performed in a 'foraging box' placed inside an electrically shielded and sound-proof chamber. The box had a wooden framework, and the walls consisted of vertical cylindrical plastic bars placed 1 cm apart from each other. The floor of the box consisted of a plastic mesh. Two foraging spouts were placed on opposite sides of the box and attached to food dispensers (Campden Instruments Ltd., USA). The dispensers were operated by custom-built Arduino hardware that was controlled by a custom-written application program in MatLab (Version 2019). On the sides of each spout, an infrared sensor pair was located to register the nose-pokes of the animals. The dimensions of the foraging box were 37 cm×26 cm x 48 cm. The distance between the spouts was 36 cm. The foraging setup for the animals is shown in Fig. 9.

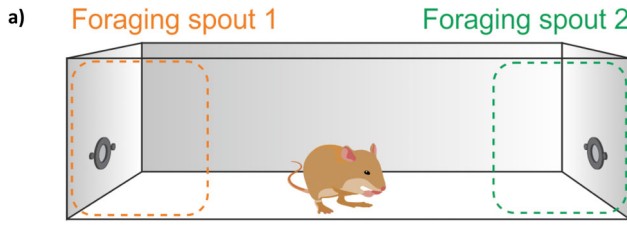

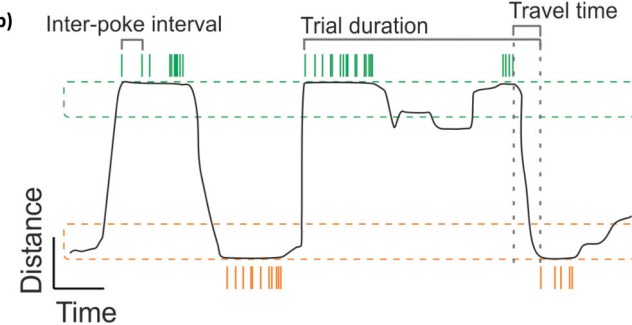

**Fig. 9 | Probabilistic foraging task in gerbils. a** Schematic illustration of the foraging task in which gerbils can access food from two 'Foraging spouts' 1 and 2, located at the opposite ends of the box. **b** Scheme indicates the animals' foraging behavior as a function of time. Color-coded circles in gray horizontal bars indicate the nose pokes at spout 1 (orange) and spout 2 (green) during an experimental session. Trial duration: A trial started with the first breaking of the light barrier by a nose poke lasting more than 100 ms (hit) at one spout and ended with a first poke at the opposite spout. The residence time per spout was defined as the duration between trial start and end. The last release of a poke in each trial marked the beginning of the following travel time that corresponds to the time until the hit poke at the opposite spout occurred.

### Probabilistic foraging paradigm

Each animal was trained once per day. In each training session, the number of trials was dependent on the animals' behavior. The total foraging time was restricted to a maximum of 30 min during the initial learning phase of five sessions. After three training sessions, the animals mastered the task more quickly, and a single experimental session was typically concluded after 15–20 min once the animal became disengaged. Each animal performed 20 sessions on 20 consecutive days. A trial was defined as the time between entering a given spout and switching to another spout. In every trial, we recorded the number and duration of nose-pokes at the spouts. Nose pokes with a duration of less than 100 ms were counted as errors and pokes lasting longer than 100 ms were recorded as hits. Error pokes remained unrewarded. Hit pokes were either rewarded with 20 mg of commercially available food pellets (Dustless precision pellets, Grain-based, 20 mg, Plexx B.V.) or unrewarded based on the current reward probability and reward outcome. The reward probabilities decreased with increasing numbers of pokes following the same exponential decay function deployed in the human visual search task (see Fig. 9). As a result, the probability of new reward capture diminished quickly, encouraging the animal to alternate between the spouts during the foraging session. Like in the human task, we used three different patch qualities (100%, 75%, and 50% reward probabilities) that were randomly interleaved between consecutive trials. To obtain more trials from the animals and to maintain the motivation of the animals, the reward probability was set to zero after the 20th hit nose-poke of a trial. After each reward, a dead time of 100 ms occurred.

### Statistics and reproducibility

We used custom-written Python code (version 3.6) for all data analyses to test whether foraging behavior in gerbils and humans was influenced by patch quality and if these influences differed between species. We employed single-factor repeated-measures ANOVA and mixed ANOVAs with Tukey HSD correction for multiple comparisons using the 'pingouin'

package. QQ-Plots checked for normality violations, using non-parametric measures like Friedman tests with Nemenyi post-hoc contrasts when necessary. Greenhouse-Geiser corrections were applied for non-sphericity. All analyses used averaged medians without outlier correction for descriptive reporting and statistical testing. Within-subject regressions were conducted using the 'lingress' function from SciPy (version 1.10.1), and Cox regressions were performed with the 'lifelines' package. The study included 42 human participants, initially 22 tested in a PC lab and subsequently 20 during neuroimaging sessions. Each human participated once. The animal study involved 18 gerbils, each performing 20 sessions over consecutive days.

### Reporting summary

Further information on research design is available in the Nature Portfolio Reporting Summary linked to this article.

### Data availability

Both animal and human data can be accessed from https://osf.io/fexgb/ [37].

### Code availability

The original custom Python code used for data analyses can be accessed from https://osf.io/fexgb/ as well as on git-hub (https://github.com/LGparrot/exploratory-attention-in-visual-foraging).

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

## Acknowledgements
The study was funded by the Deutsche Forschungsgemeinschaft, Project-ID, to S.P. and M.H., project-ID 425899996e CRC 1436 (sub-project C02). We would like to thank Gabriel Dobroschke and Sara Claassen for their help with human participants' data collection. We also thank Sabina Nowakowska for her help with the animal experiments.

## Author contributions
S.P. and M.H. conceived the project idea and secured funding. S.P. and L.G. designed the visual search task for human participants. L.G. managed the data collection for humans with the help of student assistants, wrote all Python scripts for data analysis, conducted the analysis for both human and animal data, and drafted the initial paper. F.O., M.H., and M.V. conceptualized and oversaw the animal experiments. F.O. developed and programmed the experimental setup for the rodent studies. P.S. was responsible for animal training and data collection and wrote the animal methods. All authors reviewed and commented on the final paper.

## Funding

## Competing interests
The authors declare no competing interests.
