## [Transparent Peer Review file · Communications Biology]

Differential patch-leaving behavior during probabilistic foraging in humans and gerbils

Corresponding Author: Mr Lasse Güldener

Figures originally included in the author's rebuttal have been redacted from this file.

Version 0:

Reviewer comments:

Reviewer #1

(Remarks to the Author)

The authors compare patch leaving during foraging tasks for humans and gerbils. The humans performed a multitarget visual search task where they select targets with a mouse and "fixate" (I presume the cursor hovering over the target) there for 300 ms which then gives them a monetary reward. The Gerbils on the other hand could select between two spouts which at which they had to poke their nose, which would then sporadically give a food reward. The foraging tasks that the two species performed were therefore vastly different. What was comparable between the studies was the reward schedule (or patch quality) and how it deteriorated with time. The authors investigate whether 'patch leaving' decisions for the two species are similar and whether they adhere to a few statistical principles/rules of thumb that may determine foraging and have been proposed in the literature (giving up time rule(GUT), fixed-N rule, fixed time rule, incremental rule and the marginal value theorem(MVT))

Unfortunately, I cannot recommend publication of this manuscript. The two tasks that the two species perform are simply far too different for any meaningful comparison between the two. For one, the central concept (or operational definition) of a 'patch' is very different in the two studies – and the difference is simply too large for any informative comparison. To me it seems that the authors are on the one hand studying visual attention (human study) and on the other something like a behaviouristic reward experiment, involving an unreliable schedule paradigm. It is doubtful to me whether when the gerbils switch from one spout to the other that this can in any sense be considered patch leaving. The authors say (p.22): "these results provide evidence that gerbils and humans use different patch leaving rules" and that the gerbils use a GUT rule while the humans use MVT. The tasks are so vastly different that I simply cannot accept this claim.

The bottom line here seems to be that behavior is modulated by the statistics of the environment. The authors show that this applies to vastly different species and in vastly different paradigms. But that is not a new insight and because of this I seriously doubt that the results reported here cast much light on why statistics determine behavior, how this occurs, nor what any critical differences between the two species may be.

To me it seems that the authors have two options – split these studies up and perhaps collect additional data and publish them separately, or try to design tasks that could be more meaningfully compared for the two species.

I am also not convinced by the authors conclusion that human foraging adheres to MVT. Many studies have shown that human foraging does not adhere to MVT and the authors mention some of those, but there are others (e.g. Hutchinson, Wilke & Todd, 2008; Wolfe, Cain and Aizelman, 2019; Gil-Gomez Muñoz-García, Pérez-Hernández & Wolfe, 2022; although the authors in the last one claim that their results actually do indicate this, I do not agree). What seems problematic to me is that in the literature, sometimes the findings on foraging are in line with MVT and sometimes they are not. It seems that the likeliest explanation for this is that MVT is not a principle that guides human foraging, in general. Instead, sometimes the fits to MVT may be roughly accurate, but with slight paradigm changes they do not fit. I cannot help feeling that there are so many factors influencing foraging that MVT will not work as a general explanatory principle for human foraging.

Reviewer #2

(Remarks to the Author)

This is an interesting paper describing foraging behavior in more-or-less comparable tasks in humans and gerbils. The actual foraging task is a bit odd (not bad, just odd). After you pick an item, all the items move around and that item is, then, no longer valuable, even though it still looks the same. Some other items also lose their value. The net effect is that the probability of a good selection drops fast. The main conclusion is that humans follow a marginal value theorem (MVT) patch leaving rule, while gerbils do not. The authors argue that gerbils are using a "Giving Up Time" (GUT) rule. I think that the main issues here have to do with clarity. Here are my concerns.

- 1) Did you explain why the human error bars are so much bigger than the gerbils? This is very striking in Fig 4, for example.
- 2) In Fig 5C, human variability doesn't look so bad. Why is that situation different?
- 3) I worry that the high noise in the human data explains why many statistical tests are not significant. I think you said something about that but now I can't find the comment. In any case, I am worried by the noise in the human data.
- 4) I found the discussion of the GUT rather confusing.
- 5) Oh....and what are the gray diamonds in Fig 4a & b.
- 6) Maybe it would be good to have a figure illustrating the different rules that you are testing?
- 7) MINOR: There are some odd phrases. In the Abstract "pertained exploitation" – what does it mean? On P4 "a fixation of these". There are some others, too.
- 8) MINOR: P11 says "in accordance with the German animal welfare law (insert here the protocol numbers)." I think you forgot to insert.

In summary, I would just like the paper to tell its story in a clearer manner. It does appear to be an interesting story.

Jeremy Wolfe
Signed review

Reviewer #3

(Remarks to the Author)
Summary

Foraging behavior is a fundamental necessity to animal survival across species. In this work, the authors perform a cross-species comparison of foraging strategies in humans and Gerbils. In the human version of the task, participants had to use a computer mouse to locate hidden symbolic rewards in a virtual environment and could click to progress to the next virtual environment. In the rodent task, food-deprived gerbils had to nose-poke to receive a chance of a food pellet and could physically switch between two reward locations. Both versions of the task aim to operationalize the exploration-exploitation dilemma with a temporal cost of exploration. The underlying belief is that, since the statistics of reward availability follows similar decay dynamics, cross-species differences in the resultant behavior are therefore contrastable.

The experimental design has potential for exploring inter-species differences between exploration-exploitation strategies and the limitations of extrapolating understanding of human behavior from rodent data. The major finding of this work is that there are significant qualitative differences between the human policy for selecting a new virtual environment and the gerbil policy for moving to a new food port with replenished resources. While this observation is novel and interesting, more extensive data analyses are required to support this claim.

Major Comments

- 1) The authors should explain the significance of contrasting human and rodent foraging rules. From reading the introduction, it is not clear whether there are existing theories that are being tested in these experiments regarding inter-species differences, and how the results of the study change our knowledge regarding such cognitive differences.
- 2) Also in the introduction, the authors discuss various foraging strategies and the conditions in which they could be optimal solutions to the foraging dilemma. Unfortunately, I found this important section confusing (since it appeared in an unrelated section describing the task) and lacking in rigor. I would like to see clearer, more systematic explanations of the different environment types (for example, I did not understand what is meant by "reward probabilities following a Poisson distribution"), the different patch-leaving rules, and which rule is optimal in which environment and why. This would help clarify subsequent claims, such as "Because of the unpredictable variations in the patch quality, also the GUT rule constitutes an optimal heuristic for patch-leaving decisions".
- 3) A major strength of this work is that it takes a comparative approach, studying the same computational problem in two different species. Therefore, the validity of the approach hinges on the claim that the two tasks are the same (or similar enough). However, I am not convinced that this is the case. Specifically, in the rodent version, reward probabilities decrease as a function of the number of attempts at reward in a patch, whereas in the human version the probability decreases only

after rewards. This discrepancy raises the concern that the behavioral differences may be due to the difference in task structure. For example, the last inter-capture interval seems to be more informative in the human compared to the rodent task. The authors should explain why this difference is not a likely cause for the strategy differences between the two species.

4) In figure 4, it is not clear how the data are normalized. More generally, it is hard to get a quantitative appreciation of the behavior. I would have liked to see more raw behavioral data, for example behavior during a single session, and individual subjects' data (the latter could be in supplementary). Also, I could not find information regarding the number of sessions it took to train and test the rodents, and how many trials were performed per session, what is meant in "the number of trials was dependent on the animals' behavior"?

5) Related to the previous comment, behavior in such tasks is susceptible to slow changes in behavior throughout sessions (for example, an animal will likely start a session hungry and motivated, and this motivation may decrease as it becomes satiated). However, much of the analysis in this paper assumes stationary behavior (I expect that the MCR is particularly sensitive to such variations). It is therefore important to show that the behavior was indeed stationary, or account for slow behavioral fluctuations.

6) I did not understand why residence times also included travels, shouldn't they be defined as the times between the first and last nose-poke in a patch.

7) Why is the ICR a good measure for the instantaneous collection rate? For example, if I understand correctly a trial with two rewards separated by one second followed by five seconds of futile attempts will have the same ICR as a trial with two rewards separated by one second in which the subject left immediately after the second reward. I would expect failures to also factor in the calculation of the instantaneous reward rate. On this issue, the analysis focuses on the cost of time, however reward probability depletes with the number of pokes. It will be important to show that rodent nose-pokes occur at regular intervals or account for variable poke durations.

Minor comments

1) "Lastly, the example stresses that humans and other animals have had highest chances to survive and pass their genes on to next generations if their attentional system evolved in such a way that it allows the constant balancing between exploitative and explorative behavior." – this seems like a non-sequitur.

2) "the probability of the next reward at the same spout (side of the box) decreased exponentially to zero according to a random probability." - I did not follow what it means for a probability to decay 'according to a random probability'.

3) The authors show an equation " $P(n = 1 | t_i) = Aie^{-(n-1)/5}$ " without defining what any of the variables are.

The author's response to these comments can be found at the end of this file.

Version 1:

Reviewer comments:

Reviewer #1

(Remarks to the Author)

Thanks to the authors for answering my concerns so thoroughly. I can now recommend publication.

Reviewer #2

(Remarks to the Author)

This is an extensive revision of a paper comparing foraging behavior in humans and gerbils. The biggest changes are

- the inclusion of new data
- the splitting of the data into short and long GUT (giving up time) halves.

I have just a few comments

1) My understanding is that adding subjects to an experiment is statistically risky. It encourages behavior where we add data to make results come out significant. If they were significant in the first place, we would not have added data. This is a "heads, I win. Tails, play again" strategy ... but I am not going to raise a big fuss here.

2) I worry a bit more that the new Os come from a different experiment (a different travel time (3.5 vs 5.75 sec) and a different overall duration (30 vs 60 min). It might have made more sense to do the analysis on human Exp 1 and human Exp 2, rather than on the short and long GUT split...but again I am not going to raise a big fuss here.

3) The human data still show really big variance. The data do not look very bimodal to me.

4) The presentation of the data is improved. Thank-you, for example, for Figure 1. The paper still does not tell a terribly clear story. This can be seen in the conclusion paragraph, for example. It is basically just descriptive. I would have trouble telling a good quick story about what the news here.

5) Total trivia: I happened to notice at the end, it says "The original costume python code". I imagine you mean "custom".

Reviewer #3

(Remarks to the Author)

The authors have made considerable improvements to the manuscript. They addressed my concerns and added new data, figures and analyses to support their claims. However, I still have a number of issues with the current version, particularly regarding the new analyses.

Concerns:

1) Figure 1a,b: why does the tendency to leave increase with time before patch leaving in (a) and with captures in (b). Even if the process is assumed to be somehow probabilistic, I don't understand the staircase pattern they chose (for example, why is the first jump is highest in (b)).

2) line 528: "An optimal GUT rule should account for differences in patch-quality with longer GUTs in better patches." Why? Assuming that this is not referencing a fixed GUT rule (in which case all GUTs would be the same), you might expect that the combination of decreasing GUTs with rewards, and the reward statistics would lead to shorter GUTs in better patches (in which more rewards are harvested).

3) Figure 7: what is ITT in the x labels (presumably ICI)?

4) The most serious concern I have has to do with Figures 7e,f (which I think are miss-referenced in line 560) and 8. In both cases, residence time is plotted against either GUT or number of rewards, and in both cases the slope is positive. However, I do not understand why this result is not trivial. For example, if the GUT is a part of the overall residence time then the two should naturally correlate. This is particularly important since it seems that the major conclusion that "patch-leaving decisions of both species followed an incremental mechanism" (from the abstract) relies on these figures.

5) Figure 9b: Why does the ICR increase between the last reward and patch leaving? This seems to violate the basic assumption that the patches are depleting (without which leaving doesn't make sense). An alternative is that the ICR approximation at the time of leaving is incorrect.

Author Rebuttal letter:

1

Response Letter of The Second Revision

corresponding author: Lasse GÅ¼ldener

We thank you for providing us with the opportunity for a second revision of our original manuscript, titled "When is it time to move on? Patch-leaving behavior during probabilistic foraging in humans and gerbils". Here we send you a newly revised version of the manuscript together with a point-by-point response letter, addressing the remaining comments of reviewer 2 and 3.

Reviewers' comments:

Reviewer #1 (Remarks to the Author):

Thanks to the authors for answering my concerns so thoroughly. I can now recommend publication.

Reviewer #2 (Remarks to the Author):

This is an extensive revision of a paper comparing foraging behavior in humans and gerbils. The biggest changes are

- the inclusion of new data
- the splitting of the data into short and long GUT (giving up time) halves.

I have just a few comments

1) My understanding is that adding subjects to an experiment is statistically risky. It encourages behavior where we add data to make results come out significant. If they were significant in the first place, we would not have added data. This is a "heads, I win. Tails, play again" strategy but I am not going to raise a big fuss here.

Response:

We agree that merely adding subjects to an existing study is statistically risky especially when frequentist statistical tools are used. However, we conducted the fMRI experiment as a second follow-up study and having the two datasets of behavioral data, we considered that combining both would result in a stronger case. Importantly, we changed central parts of the analysis and ran new analyses instead of just rerunning previous ones just with a larger sample size. We would like to mention that the expansion of the dataset has not resulted in any changes to the overall findings or the statistical results.

2) I worry a bit more that the new Os come from a different experiment (a different travel time (3.5 vs 5.75 sec) and a different overall duration (30 vs 60 min). It might have made more sense

to do the analysis on human Exp 1 and human Exp 2, rather than on the short and long GUT split; but again I am not going to raise a big fuss here.

Response:

We acknowledge the difference in travel time and overall duration between the two sets of observations as limiting factors. However, the difference in travel time with longer travel times in the fMRI experiment due to technical reasons, was still rather small compared to other experiments (e.g., Wolfe, 2013, experiment 3) that focused on manipulating travel times directly. Importantly, within the short- and long-GUT groups, there were no differences in GUTs between fMRI and PC-lab participants. Thus, although we acknowledge the potential impact of the differences in travel time and overall duration on the timing of the observed patch-leaving decisions, we believe that the split based on GUTs was still justifiable.

3) The human data still show really big variance. The data do not look very bimodal to me.

Response:

We agree that the data of the human GUTs showed a rather evenly distribution with large variance. Importantly, the high-GUT vs. low-GUT groups were introduced for data analytic reasons, i.e., to compare two extreme groups. We do not claim that these two groups are "natural" subpopulations resulting from the observed variance structure. We now mention this in the discussion, p. 27, l. 1016 - 1035:

"High-GUT and low-GUT groups did not represent natural subpopulations

Lastly it needs to be noted that the high-GUT vs. low-GUT groups were introduced for data analytic reasons, i.e., to compare two extreme groups. Importantly, these two groups were no "natural" subpopulations resulting from the observed variance structure. The group splitting was rather a means to deal with the large variance in humans' residence as well as giving-up times data. Human subjects received only a minimum of training to familiarize them with the task. Thus, when starting the experiment their performance may had not yet stabilized introducing more within-subject noise compared to the intensively trained and thus more consistently behaving animals. Additionally, inter-individual differences in the propensity to explore versus exploit could have been driving the large dispersion in participants' GUTs and residence times. Even though the optimal strategy was to leave patches rather early, subjects' disposition to either persist or to leave more readily certainly impacted their behavior and introduced the large dispersion in both parameters. In support of this notion, a large study in humans recently confirmed that ADHD symptom-like behavior assessed by self-report predicted the residence times during a patchy virtual foraging task (Barack et al., 2024). The higher a subject scored on a ADHD self-report screening assessment, the shorter their residences times were. This finding not only offers an interesting perspective on the evolutionary benefits that ADHD traits may offer, it also provides an explanation why inter-

individual differences in strategies during a virtual foraging task may be large, especially if participants had not undergone extensive training before.

4) The presentation of the data is improved. Thank-you, for example, for Figure 1. The paper still does not tell a terribly clear story. This can be seen in the conclusion paragraph, for example. It is basically just descriptive. I would have trouble telling a good quick story about what the news here.

Response:

We revised the conclusions. It now more clearly conveys a concise 'take-home' message for the reader, p. 28, l. 1056 - 1072:

The findings indicate that rodents and humans share similar decision rules in environments with comparable reward unpredictability. Both species adapt to changing reward probabilities by avoiding disadvantageous rules of thumb for patch-leaving, using novel reward encounters to extend residence times. However, a subset of humans exhibited overharvesting, leading to prolonged GUTs and reduced earnings, a maladaptation that is lacking/not observed in rodents. Thus, optimally-timed patch-leaving was observed in non-overharvesting humans and gerbils, aligning with the MVT. These results underscore the value of inter-species comparisons but also highlight the variability in human foraging decisions, suggesting that individual differences play a role in adherence to ecological models. Altogether, this study paves the way for future research comparing the neural substrates involved in exploratory decision-making between species, contributing to our understanding of the phylogenesis of cognitive control structures in the brain.

5) Total trivia: I happened to notice at the end, it says 'The original costume Python code'. I imagine you mean 'custom'.

Response: We corrected the typo accordingly.

Reviewer #3 (Remarks to the Author):

The authors have made considerable improvements to the manuscript. They addressed my concerns and added new data, figures and analyses to support their claims. However, I still have a number of issues with the current version, particularly regarding the new analyses.

Concerns:

1) Figure 1a,b: why does the tendency to leave increase with time before patch leaving in (a) and with captures in (b). Even if the process is assumed to be somehow probabilistic, I don't understand the staircase pattern they chose (for example, why is the first jump is highest in (b)).

4

Response:

The new Figure 1 b) was indeed misleading: consistent with the fixed-n rule each target encounter should equally increase the probability to leave the patch. Thus, we revised the Figure accordingly so that the steps are now equal, p. 6, l. 224:

2) line 528: "An optimal GUT rule should account for differences in patch-quality with longer GUTs in better patches." Why? Assuming that this is not referencing a fixed GUT rule (in which case all GUTs would be the same), you might expect that the combination of decreasing GUTs with rewards, and the reward statistics would lead to shorter GUTs in better patches (in which more rewards are harvested).

Response:

Our hypothesis of longer GUTs in high-quality patches (i.e., increases in GUTs with increasing patch-quality) stems from McNair's article 'Optimal giving-up times' (1982):

p. 513

I show in the next section that, based on a model which is entirely analogous to Charnov's but

which is designed to make predictions concerning GUT's, larger GUT's should be used in better
5

patches. This prediction seems commonsensical, suggesting a forager should be more persistent
in patches it knows are better.â

p. 523

âAn important result obtained above is that the optimal GUT should not be the same in patches
of different quality within the same habitat. Rather, larger GUT's should be used in better
quality patches. If, as Croze (1970) suggests, the GUT is viewed as a measure of a forager's
persistence in searching for more food in a patch, then theory predicts that a forager should be
more persistent in better quality patches.â

3) Figure 7: what is ITT in the x labels (presumably ICI)?

Response

ITT refers to the inter-target time, i.e., the time interval between two consecutive reward
captures, and is, thus, equal to the ICI, i.e., the inter-capture interval. For more consistency
between text and Figure, now only the ITT term is used throughout the manuscript.

4) The most serious concern I have has to do with Figures 7e,f (which I think are miss-referenced in line
560) and 8. In both cases, residence time is plotted against either GUT or number of rewards, and in both
cases the slope is positive. However, I do not understand why this result is not trivial. For example, if the
GUT is a part of the overall residence time then the two should naturally correlate. This is particularly
important since it seems that the major conclusion that "patch-leaving decisions of both species followed an
incremental mechanism" (from the abstract) relies on these figures.

Response

The predictor and the outcome were indeed mixed up in this analysis. The revised Figure 8 (see
below) now shows the incremental relationship between reward captures and residence times.
Each additional reward encounter increased the subject's probability to stay in the current patch
consistent with the incremental rule of patch-leaving (see Figure 1 C). This notion was further
supported by the cox-regression showing that target captures had a protective effect on the
âhazardâ of patch-leaving. We corrected the respective part of the results section, p. 18, l. 639
â 643:

âIndividual slopes in both subgroups of human participants were in all cases positive
and on average significantly above zero, [short-GUT: mean slope = 3.372 ± 0.975 , $t(20)$
= 15.457, $p < .001$, long-GUT: mean slope = 3.748 ± 0.712 , $t(20)$ = 23.528, $p < .0001$] (see
Figure 8 a). This suggested that participants indeed extended their residence time in response
to a new reward capture, consistent with the incremental patch-leaving rule. Intriguingly,
within-subject slopes in the short-GUT group of humans, that had shown less evidence for a
6

GUT rule, did not differ from the slopes of the long-GUT group, $t(40) = -1.390$, $p = .172$,
indicating that the incremental effect of reward captures on the likelihood to stay in the current
patch was comparable in both groups of humans. We conducted the same within-subject
regressions also for the short- and long-GUT group of gerbils and obtained a similar pattern
of results. The mean slope was 3.368 ± 0.690 in the short-GUT group, $t(8) = 13.795$, $p < .0001$,
and 4.507 ± 0.674 in the long-GUT group, $t(8)=18.900$, $p < .0001$. The average slopes of long-
GUT gerbils were significantly larger than the average slope of the short-GUT gerbils,
indicating stronger incremental relationship in those animals with longer GUTs, $t(16) = 3.335$,
 $p = .004$ (Figure 8 b).â

We also revised Figure 8 accordingly, p. 18, l. 661:

The code error in Figure 7e and f was also present in the sunk cost analysis. We also re-evaluated the correlation between GUTs and overall residence times, considering GUTs as part of the residence time. The revised analysis aimed to test for an incremental relationship between time spent up to the onset of the GUT (i.e., residence time - GUT) and the following GUT. However, this analysis did not show evidence for a sunk-cost effect and was consequently omitted from the manuscript. Importantly, this analysis was not central to our main conclusions, whereas the within-subject regressions of residence times on the number of reward captures, which maintained consistent results after the error correction, are pivotal to our key finding that both species' patch-leaving aligns with the incremental rule.

5) Figure 9b: Why does the ICR increase between the last reward and patch leaving? This seems to violate the basic assumption that the patches are depleting (without which leaving doesn't make sense). An alternative is that the ICR approximation at the time of leaving is incorrect.

Response

7

The instantaneous collection rate at patch-leaving is given by the inverse of a subject's GUT. A short GUT (e.g., 3 seconds) results thus in an estimated collection rate of 0.3, while a longer GUT (e.g., 6 seconds) yields 0.16. Importantly, this estimate is independent of actual reward encounters, relying solely on the participant's propensity to continue foraging unsuccessfully. A lower GUT implies a higher estimated collection rate at patch-leaving, potentially even leading to an increase of the rate compared to the collection rate at which the preceding rewards had been captured, as seen in Figure 9b at t_{last} and t_{-1} on the x-axis. It is essential to note that this estimate does not originate from an actual reward encounter and, thus, does not violate the assumption of a depleting patch.

We added parts of this explanation to the Figure 9's caption, p. 20, l. 716 - 722:

Importantly, this estimate of collection rate in the moment of patch-leaving ($= 1/GUT$) is independent of actual reward encounters, relying solely on the participant's propensity to continue foraging unsuccessfully. A lower GUT implies a higher estimated collection rate at patch-leaving, potentially even leading to an increase of the rate compared to the collection rate at which the preceding rewards had been captured, as seen in Figure 9b where the estimated CR at patch-leaving was higher than the CR of the last and second last reward (i.e., t_{-1} on the x-axis). It is essential to note that this estimate does not originate from an actual reward encounter and, thus, does not violate the assumption of a depleting patch.

Version 2:

Reviewer comments:

Reviewer #2

(Remarks to the Author)

I am not entirely sure, on the basis of these data, that I am entirely convinced "that humans and rodents use similar heuristics to time their patch-leaving behavior." However, I have no further specific comments. Thank-you for your revisions.

Jeremy Wolfe

Reviewer #3

(Remarks to the Author)

I appreciate the significant improvements the authors made to the manuscript.

However, I still do not understand the analyses shown in figure 7 and 8 or the authors reply to my previous comment 4. I'll repeat it briefly here: Residence time and number of rewards per trial are correlated by design, and would be even if leaving decisions were completely random. The statement that "participants indeed extended their residence time in response to a new reward capture" seems to me like a case in which causation is inferred from correlation. Also, I couldn't find Figure 7e.f.

Author Rebuttal letter:

Response Letter

corresponding author: Lasse Gårdner

Dear reviewers,

after receiving the constructive feedback provided by the reviewers on our manuscript, titled "When is it time to move on? Patch-leaving behavior during probabilistic foraging in humans and gerbils", we are pleased to send you a revised version of the manuscript together with a brief response letter, addressing the last two remaining concerns. In the following we address each of your remarks in detail.

Reviewers' comments:

Reviewer #2 (Remarks to the Author):

I am not entirely sure, on the basis of these data, that I am entirely convinced "that humans and rodents use similar heuristics to time their patch-leaving behavior." However, I have no further specific comments. Thank-you for your revisions.

Response:

We removed the statement in question from the abstract and we thank you for your revisions you provided to improve our manuscript.

Reviewer #3 (Remarks to the Author):

I appreciate the significant improvements the authors made to the manuscript.

However, I still do not understand the analyses shown in figure 7 and 8 or the authors reply to my previous comment 4. I'll repeat it briefly here: Residence time and number of rewards per trial are correlated by design, and would be even if leaving decisions were completely random. The statement that "participants indeed extended their residence time in response to a new reward capture" seems to me like a case in which causation is inferred from correlation. Also, I couldn't find Figure 7e.f.

Response:

Thank you for the continued dialogue and the opportunity to further clarify our analyses. To address the remaining concern we conducted a simulation-based analysis following the analytical approach shown in previous studies (e.g., Hutchinson et al., 2008). The key idea was that if we simulated the number of obtained rewards per patch assuming a randomly proportional relationship between residence times and number of obtained rewards and performed the same within-subject data on the simulated data, the resulting slopes should be smaller than the observed slopes obtained from analyzing the real data if our hypothesis of an incremental (i.e., rather causal) relationship between the two variables was true. We added the following paragraph to the data analysis on p. 18-19, l. 611-686:

Individual slopes in both subgroups of human participants were in all cases positive and on average significantly above zero, [short-GUT: mean slope = 3.372 ± 0.975 , $t(20) = 15.457$, $p < .001$, long-GUT: mean slope = 3.748 ± 0.712 , $t(20) = 23.528$, $p < .0001$] (Figure 8 a). This suggested that participants indeed extended their residence time in response to a new reward capture, consistent with the incremental patch-leaving rule. Within-subject slopes in the short-GUT group of humans, that had shown less evidence for a GUT rule, did not differ from the slopes of the long-GUT group, $t(40) = -1.390$, $p = .172$, indicating that the incremental effect of reward captures on the likelihood to stay in the current patch was comparable in both groups of humans. We conducted the same within-subject regressions also for the short- and long-GUT group of gerbils and obtained a similar pattern of results: the mean slope was 3.368 ± 0.690 in the short-GUT group, $t(8) = 13.795$, $p < .0001$, and 4.507 ± 0.674 in the long-GUT group, $t(8) = 18.900$, $p < .0001$. The average slopes of long-GUT gerbils were significantly larger than the average slope of the short-GUT gerbils, indicating stronger incremental relationship in those animals with longer GUTs, $t(16) = 3.335$, $p = .004$ (Figure 8 b).

Although we anticipated that reward encounters affected residence times incrementally, we agree that one must be careful to not infer causation from correlation: regressing residence times on the number of rewards (residence times $\sim 1 + \text{reward encounter} * \hat{\beta}$), we anticipated that the slopes would be positive, reflecting an incremental mechanism between the two variables. Yet, alternatively it could be that longer residence times might simply increase the likelihood of finding more rewards, thereby creating a positive correlation between captures and time spent. In this scenario the time spent in a patch is random and not influenced by capture success. To test this "random model" against our incremental model, we followed Hutchinson's approach (2008) and took individuals' residence times as well as the initial reward probabilities and simulated the number of reward encounters on a trial-by-trial basis, assuming that reward captures are simply proportionally related to the residence times (number of reward captures = residence time * initial reward probability / 100). In a second step, we conducted the same within-subject regressions, regressing residence times on the number of (simulated) reward captures on a trial-by-trial basis. Lastly, we compared the slopes obtained in the simulation (i.e., the increment in residence times by a reward encounter if the relationship was random) with the slopes observed in the real data. The results indicated significantly larger slopes in the real data, supporting the idea that participants indeed extended their residence time following each capture (Hutchinson et al., 2008; Mata et al., 2009).

Simulations of reward encounters and the within-subject regressions were carried out 100 times for each subject. We then recombined the resulting slopes via bootstrapping on group-level using 100000 bootstraps to get an estimate of the distribution of slopes on group-level that would be obtained assuming that the time spent in a patch is random and not influenced by capture success. Because there was no indication for differences between long- and short-GUT humans, we carried out this control analysis jointly for both subgroups. The results are shown in Figure 8 a) and confirmed a smaller slope (mean slope = 2.138 ± 0.326 , 95% bootstrap interval [2.137, 2.159]) obtained from within-subject regressions on simulated data in which the relationship between residence times and number of reward encounters was randomly proportional, $t(40) = -11.061$, $p < .001$.

The same analysis we also conducted for the gerbil data. Here, however, the additional analysis provided no further support for an incremental relationship between reward encounters and residence times (Figure 8 b), and the slope obtained from analyzing the simulated data (mean slope = 5.475 ± 0.252 , 95% bootstrap interval [5.453, 5.480]) was even larger than the slopes based on the observed data, short-GUT gerbils $t(25) = 8.861$, $p = .999$, long-GUT gerbils $t(25) = 4.199$, $p = .999$.

Taken together, in support of the incremental rule, within-subject regressions in both species showed that individuals extended their residence times incrementally following a new reward capture. However, only in humans these slopes exceeded a simulation-based slope, assuming a randomly proportional relationship. Thus, analyzing the association between reward encounters and residence times provided evidence for the incremental rule of patch-leaving (Figure 1 c) only in humans.

Regarding the confusion concerning Figure 7 e) and Figure f), as mentioned in the previous revision, the sunk-cost analysis originally shown in these panels has been removed from the manuscript. Consequently, Figure 7 now includes only panels a) through d). Additionally, we have incorporated a new analysis of the slopes obtained from the data simulation into Figure 8, as shown below.

Version 3:

Reviewer comments:

Reviewer #3

(Remarks to the Author)

I think that the new analysis (figure 8) does address my concern, but am confused regarding its interpretation. If the finding is that regression slopes exceed chance values only in humans, why do the authors still claim that "both species followed an incremental mechanism"?

Author Rebuttal letter:

Response Letter

corresponding author: Lasse Gårdner

Dear reviewer,

we thank you for your final comments on our manuscript, titled "When is it time to move on? Patch-leaving behavior during probabilistic foraging in humans and gerbils", send you a revised version of the manuscript in which we toned down our conclusion so that they are now in full alignment with the results.

Comment #1

In particular, I agree with the remaining statement of Reviewer #3 that the descriptions and summaries of the simulation results seem inconsistent. On one hand, the simulation analysis clearly shows that only in humans, not gerbils, did the data refute the null-hypothesis. This is clearly acknowledged in the description of the results -- e.g., "in gerbil, unfortunately, the correlations may be spuriously produced by the association between time and reward that is inherent in the task.

And yet, other statements continue to claim that both species follow an incremental rule -- i.e., interpret the correlations in gerbils to imply a causal relation between more rewards and longer dwell times. One example is in the new text for Fig. 8, which states: "Taken together, in support of the incremental rule, within-subject regressions in both species showed that individuals extended their residence times incrementally following a new reward capture.

Please go through the entire manuscript and tone down the conclusions to make them consistent with the results. Please highlight all changes in the manuscript text file so we can verify them before making a final decision."

Response:

We apologize for the inconsistency and the lack of clarity in the last version and have now removed all statements that continued to claim that both species follow an incremental rule the results section as well as from the discussion, where we now highlighted that the results support this claim only in humans.

Results, p. 19, l. 660 - 601:

'Taken together, analyzing the association between reward encounters and residence times provided evidence for the incremental rule of patch-leaving (Figure 1 c) only in humans.'

Discussion, p. 24, l. 840:

âReward captures incrementally delay patch-leaving only in humans.

Due to the probabilistic decay function and the additionally varying initial probabilities, predicting the reward probabilities for both humans and gerbils was exceedingly challenging. To tackle single reward encounters provided a first means for our gerbils and humans to learn about the quality of the current patch and each new reward encounter could be perceived as an indication that the current patch may be of good quality, and, thus worthwhile to spend more time in it. Consistent with this notion, the within-subject regressions of residence times on the number of reward captures, showed a positive slope in all individuals of both species, suggesting that residence times were extended incrementally following a new reward capture (Iwasa, 1981). However, longer residence times could have simply increased the likelihood of finding more rewards, resulting in a positive correlation between captures and time spent. In this case the time spent in a patch is random and not causally driven by capture success. An additional simulation analysis supported a causal relationship (i.e., the incremental rule) only in humans but not in gerbils. The slopes from data simulated with the assumption of a randomly proportional relationship between reward encounters and residence times were less steep than those slopes obtain from within-subject regressions on the actual data, suggesting that the human participants did indeed increase time in a patch with each reward capture. In gerbils, an analogous simulation analysis did not confirm this notion. Our findings in humans confirm previous findings (e.g., Wilke et al., 2009), whereas the findings in our gerbils are inconsistent with the results in mice by Lottem and colleagues (2018). The authors analyzed mice's nose-pokes and fitted their data with a proportional hazard model. The estimated hazard rate reflected the probability to leave a

current patch as a function of nose-pokes that started at its minimum with the beginning of a trial and would increase with each unrewarded nose poke. Each rewarded nose poke, however, decreased the hazard of leaving prolonging residence times, similar to our results in the cox regression, where new reward encounters decreased the 'risk' of leaving. While the mice collection rates at the time of leaving were in keeping with the MVT, the incremental model showed a significantly better fit compared to a MVT-based model fitting in Lottem's study. Thus, using the same foraging paradigm in gerbils, it will be important to test if those findings can be replicated when analyzing nose-pokes instead of residence times and reward encounters. At this point, only our human data confirms with the incremental rule driving patch-leaving decisions.

Comment #2

Even more disturbingly, this causal conclusion is highlighted in the abstract, which states that both species "followed an incremental mechanism based on reward encounters [...] that is considered optimal for maximizing reward yields in variable foraging environments".

Response:

We removed the inconsistent claim also from the abstract that now reads as follows, p 1, l. 30-41:

Foraging confronts animals, including humans, with the need to balance exploration and exploitation: exploiting a resource until it depletes and then deciding when to move to a new location for more resources. Research across various species has identified rules for when to leave a depleting patch, influenced by environmental factors like patch quality. Here we compared human and gerbil patch-leaving behavior through two analogous tasks: a visual search for humans and a physical foraging task for gerbils, both involving patches with randomly varying initial rewards that decreased exponentially. Patch-leaving decisions of humans but not gerbils followed an incremental mechanism based on reward encounters that is considered optimal for maximizing reward yields in variable foraging environments. The two species also differed in their giving-up times, and some human subjects tended to overharvest. However, gerbils and individual humans who did not overharvest were equally sensitive to declining collection rates in accordance with the marginal value theorem. Altogether this study introduces a novel paradigm for a between-species comparison on how to resolve the exploitation-exploration dilemma.

Again, we apologize for the inconsistencies and thank you for your time and effort in this review process.

Sincerely yours,

Lasse Waldener, on the behalf of all contributing authors.

Response Letter

corresponding author: Lasse Güldener

to

Jacqueline Gottlieb, PhD
Editorial Board Member
Communications Biology
orcid.org/0000-0001-6507-4375

After receiving the constructive feedback provided by the reviewers on our manuscript, titled "*When is it time to move on? Patch-leaving behavior during probabilistic foraging in humans and gerbils*", we are pleased to send you a revised version of the manuscript together with a comprehensive response letter. It is worth noting that the feedback from reviewer 1 and 3 was conflicting. Reviewer 1 did not recommend a comparative approach in general, while Reviewer 3 advocated for it. In response, we have made substantial improvements to the manuscript, which include a more comprehensive discussion of why the chosen approaches were adopted and a more careful exploration of their comparability. Furthermore, in line with the concerns of Reviewer 2 regarding the high variability in the human data introducing a concern regarding the statistical power, we included additional data in both humans (+ 20) and gerbils (+10) and conducted entirely new data analyzes to ensure the robustness and validity of the statistical results. In the following we address each reviewer's remarks in detail.

Reviewers' comments:

Reviewer #1 (Remarks to the Author):

The authors compare patch leaving during foraging tasks for humans and gerbils. The humans performed a multitarget visual search task where they select targets with a mouse and "fixate" (I presume the cursor hovering over the target) there for 300 ms which then gives them a monetary reward. The Gerbils on the other hand could select between two spouts which at which they had to poke their nose, which would then sporadically give a food reward.

The foraging tasks that the two species performed were therefore vastly different. What was comparable between the studies was the reward schedule (or patch quality) and how it deteriorated with time. The authors investigate whether 'patch leaving' decisions for the two species are similar and whether they adhere to a few statistical principles/rules of thumb that may determine foraging and have been proposed in the literature (giving up time rule(GUT), fixed-N rule, fixed time rule, incremental rule and the marginal value theorem (MVT)

1) Unfortunately, I cannot recommend publication of this manuscript. The two tasks that the two species perform are simply far too different for any meaningful comparison between the two. For one, the central concept (or operational definition) of a 'patch' is very different in the two studies – and the difference is simply too large for any informative comparison.

To me it seems that the authors are on the one hand studying visual attention (human study) and on the other something like a behavioural reward experiment, involving an unreliable schedule paradigm. It is doubtful to me whether when the gerbils switch from one spout to the other that this can in any sense be considered patch leaving.

Response

Independent of species, modality, etc. exploration/exploitation dilemmas have been described across the animal kingdom. The paradigm we used for the gerbils was introduced by Lottem et al. 2018 to study the role of 5-HT during foraging in mice ('Activation of serotonin neurons promotes active persistence in a probabilistic foraging task, doi: 10.1038/s41467-018-03438-y'). The idea of these authors was that mice had to learn the statistics of the environment and infer when to leave a depleted foraging site for the next. Mice therefore had to actively nose-poke in order to exploit a given site. This task thereby is matching the principles of *probabilistic patch-like structured foraging environment* in which foragers constantly have to decide how long they should exploit a given patch and when to explore a new one as the current patch of unknown quality continues to deplete with ongoing exploitation. Concerning the human paradigm, we build on the literature showing that the visual search paradigm can be used to investigate foraging behavior in humans (Kristjánsson et al., 2020; Wolfe, 2013). Although gambling tasks have been dominating the literature concerned with the exploitation-exploration dilemma (e.g., Daw et al., 2006), there is good reason to use serial visual search tasks because they require *serial* decision-making. Such task demand captures a real-life foraging scenario in patchy environments (e.g., berry picking, Wolfe, 2013) that entails continuous patch-leaving decisions much better than gambling tasks in which a choice of simultaneous options needs to be made. We elaborated this point in the revised version of the introduction to support our choice of the experimental tasks, **p. 2-3, 1.76-101**:

'The human participants performed a visual foraging task embedded in a visual search paradigm (e.g., Kristjánsson et al., 2020; Wolfe, 2013). Animals, including our hunter-gatherer ancestors, often encountered foraging environments with spatially and temporally distributed patches (e.g., forest districts with varying prey richness at different locations and distances within one habitat). Foraging in such environments involves serial decision-makings, incurring temporal travel costs as animals move from one patch to another. It is this very aspect of foraging that is better captured in a serial visual search task compared to the bandit-like gambling tasks, traditionally used to study the exploration-exploitation dilemma (e.g., Daw et al., 2006; Laureiro-Martínez et al., 2015). In the latter, decisions involve simultaneous choices. In our visual search task, the participants searched on monitor displays for target items among distractors and had to decide whether to continue searching in the current display (i.e., patch) or to switch to a new display with novel targets by pressing a button (patch-leaving). In each display, participants earned a monetary reward each time they located a target item using the PC-mouse. This approach resonates with previous work on foraging behavior in humans that also combined a serial visual search paradigm with a patch-based foraging task (Kristjánsson et al., 2020; Wolfe, 2013). Experimental approaches that work well in humans can be much

more difficult to apply in rodents. Thus, to introduce adequate task conditions for patch-based foraging for the gerbils, we adopted a foraging task established by Lottem et al. (2018). Here the foraging setup consisted of a box-like arena with two foraging spouts located on opposite sides of the box. The animals were trained to nose-poke (forage) at one of two spouts that dispensed food rewards). Importantly, both the human and the rodent task were equivalent in their probabilistic as well as in their patch-based structure: reward probabilities followed the same exponentially decreasing function in both paradigms so that the foraging success declined the longer the foraging humans and animals remained in the same patch. Thus, both animal and human subjects were comparably forced to make patch-leaving decisions to achieve optimal foraging.'

In addition, given the additional data and new analyzes, the new manuscript focuses much more on the *commonalities* in foraging behavior between the two species instead of trying to stress differences between species. This resonates with the central message that we aim to convey in this manuscript, namely that despite obvious differences in the two tasks, we are able to show *similarities* in how humans and rodents time their decisions to switch from exploitation to exploration, see discussion, p.22-24, l. 782-826:

'Humans and gerbils both adapt to changing reward probabilities.

Timing patch departures based on a fixed number of reward captures or based on a fixed amount of time only works well if the forager roams an environment that offers patches that do not differ greatly in quality (see Wilke, 2004). However, both the fixed N- as well as the fixed-T rule are not optimal if patches within an environment differ greatly in quality (Iwasa et al., 1981; Stephens & Krebs, 1987). Thus, given the randomly changing reward probabilities in our paradigms, the use of these rules would have been disadvantageous for our species. The findings that both residence times and reward captures increase with increasing patch quality, confirmed this assumption. However, humans showed high variability in individual residence times and GUTs. We therefore divided the sample of humans based on the median GUTs into long- and short-GUT individuals to evaluate whether different rules apply to these two subgroups of human participants. For better comparison we did the same also for the group of gerbils. Both subgroups of both species showed evidence for the incremental rule of patch-leaving (see Figure 1 c)) and all animal and human subjects were sensitive to sunk costs. However, only the short-GUT group of humans but all gerbils timed their patch-leaving optimally according to the MVT. Lastly, a cox-regression suggested that both species used similar cues to time their patch-leaving.

Reward captures incrementally delay patch-leaving in both species

Due to the probabilistic decay function and the additionally varying initial probabilities, predicting the reward probabilities for both humans and gerbils was exceedingly challenging. To tackle single reward encounters provided a first means for our gerbils and humans to learn about the quality of the current patch and each new reward encounter could be perceived as an indication that the current patch may be of good quality and thus worthwhile to spend more

time in it. Consistent with this notion, the within-subject regressions, where we regressed residence times on the number of reward captures, showed a positive slope in all individuals of both species, demonstrating that residence times were extended incrementally following a new reward capture. This led to significantly longer residence times in high compared to low quality patches in both species and is consistent with an incremental mechanism [VM1] driven by reward encounters that incrementally increase residence times by postponing the patch-leaving (Iwasa, 1981). Further support of this notion was given by the results of the cox regression in both species showing that the number of reward captures had a significant protective effect on the ‘risk’ of patch-leaving (i.e., delaying patch-leaving). This result confirms previous findings in humans (e.g., Wilke et al., 2009) and agrees with the results in mice by Lottem and colleagues (2018), from whom we adopted the probabilistic foraging task for our gerbils. However, in contrast to our analysis, Lottem et al. (2018) analyzed mice nose-pokes and fitted these data with a proportional hazard model. The estimated hazard rate reflected the probability to leave a current patch as a function of nose-pokes that started at its minimum with the beginning of a trial and would increase with each unrewarded nose poke. Each rewarded nose poke, however, decreased the hazard of leaving prolonging residence times. While the mice collection rates at the time of leaving were in keeping with the MVT, the incremental model showed a significantly better fit compared to a MVT-based model fitting. Using the same foraging paradigm in gerbils, but analyzing reward captures and residence times instead of nose-pokes, we can replicate this finding for gerbils. By designing a task with a foraging environment that has a comparable reward structure, we can also show that human foragers similarly adopt an incremental mechanism for patch-leaving. This demonstrates that both species adapt their foraging strategy in a similar way when facing an environment of unknown and variable reward structure, where some patches offer more rewards than others.

and see also, p. 24-25, l. 866-888:

‘Evidence for the sunk-cost effect in both species

Humans and gerbils tended to extend their GUTs depending on the amount of time they had already spent in the current display. This was confirmed by the results of the within-subject regressions between giving-up times and residence times. Here, all but four humans had positive slopes which is consistent with the sunk-cost effect (also referred to as the ‘Concorde fallacy’; e.g., Arkes & Ayton, 1999). This effect reflects a cognitive bias manifested as the inclination of individuals to persist in allocating resources to an ongoing endeavor or decision that has already accumulated substantial costs (referred to as sunk costs), even when the prospect of success is low or the initial choice was mistaken. Since only additional costs and benefits incurred by continuing with a particular course of action (i.e., marginal), but not past costs (i.e., in our case the residence time that the subject had already invested in foraging in the current patch), should factor into the decision making, the effect is considered maladaptive (Navarro & Fantino, 2005). Interestingly, we observed a parallel pattern of outcomes among gerbils. The presence of positive individual slopes indicated that these animals were also attuned to their prior time investments. This sensitivity was evidenced by the tendency of gerbils to tolerate longer durations of unsuccessful search as their cumulative time in the current patch

increased, before they left the patch. Contrary to the initial assumption that only humans succumb to the sunk-cost effect (Arkes & Ayton, 1999), mounting evidence points towards its presence in diverse species, including rodents like mice and rats (Redish et al., 2022; Sweis et al., 2018b; Wikenheiser, Stephens & Redish, 2013), as well as avians and other creatures (for an overview, see Mahalgães & White, 2016; Pattison, Zental & Watanabe, 2012; Watzek & Brosnan, 2020). Our findings align with this growing body of evidence, providing further support to the notion that the sunk cost phenomenon extends across a range of species.'

2) The authors say (p.22): "these results provide evidence that gerbils and humans use different patch leaving rules" and that the gerbils use a GUT rule while the humans use MVT. The tasks are so vastly different that I simply cannot accept this claim. The bottom line here seems to be that behavior is modulated by the statistics of the environment. The authors show that this applies to vastly different species and in vastly different paradigms. But that is not a new insight and because of this I seriously doubt that the results reported here cast much light on why statistics determine behavior, how this occurs, nor what any critical differences between the two species may be.

Response

The key goal was to also identify commonalities between the two species but not to only shed light on potential differences. New evidence from within-subject regressions showed similarities in that both species used single target encounters as cues that the current patch was of high quality, leading to an incremental relationship between single reward encounters and residence times, results section **p. 17-18, l. 597-640**:

'Increments in residence times following reward captures in both species

So far, the data had provided good evidence for the GUT rule in both groups of gerbils as well as in the long-GUT human subjects. The short-GUT group of humans, however, showed data that were inconsistent with such a rule. Thus, we next examined the relationship between reward captures within a patch and residence times. Given the unpredictable changes of patch-quality, rewards encountered within a patch provide the only viable estimate of the underlying patch-quality. If foragers relied on this estimate, they would extend their residence times incrementally with each novel reward capture because each new reward encounter would suggest that the current patch is potentially of high quality. To test this, we again calculated within-subject regressions but this time we regressed the residence times on the number of reward captures (Hutchinson, Wilke & Todd, 2008; Mata, Wilke & Czienskowski, 2009; Wilke, Gigerenzer & Jacobs, 2006). This way, we obtained a slope and intercept for each participant, where the intercept represented the initial time spent in the current display without a reward detection, and the slope represented the increase in the residence time with each new reward capture.

Individual slopes in both subgroups of human participants were in all cases positive and on average significantly above zero, [short-GUT: mean slope = 0.235 ± 0.072 , $t(20)$

$= 14.643, p < .001$, long-GUT: mean slope $= 0.156 \pm 0.050, t(20) = 13.965, p < .001$] (see Figure 8 a). This suggested that participants indeed extended their residence time in response to a new reward capture, consistent with the incremental patch-leaving rule. Intriguingly, within-subject slopes in the short-GUT group of humans, that had shown less evidence for a GUT rule, were significantly steeper compared to the slopes of the long-GUT group, $t(40) = 4.179, p < .001$, indicating that the incremental effect of reward captures on the likelihood to stay in the current patch was stronger in short-GUT humans. We conducted the same within-subject regressions also for the short- and long-GUT group of gerbils and obtained a similar pattern of results. The mean slope was 0.0790 ± 0.048 in the short-GUT group, $t(8) = 4.154, p = .003$, and 0.095 ± 0.061 in the long-GUT group, $t(8) = 4.394, p = .002$. The average slopes did not differ between the two groups, $t(16) = -0.934, p = .364$ (Figure 8 b).

Taken together, in support of the incremental rule, within-subject regressions in both species showed that individuals extended their residence times incrementally following a new reward capture. Intriguingly, this incremental effect of reward capture on residence time was stronger in the human short-GUT group that had shown less evidence for a consistent GUT rule compared to the human long-GUT group. The same trend was observed in gerbils, yet lacking statistical significance.

Fig. 8. Incremental relationship between reward captures and residence times. a) Regression plot on group level for all humans. Black dots show individual data points. The blue line shows the averaged regression line of the individual intercepts and slopes obtained from the within-subject regressions regressing residence times on the number of rewards for short-GUT humans. The green line shows the same for the long-GUT humans. The mean individual slope was significantly higher in the short- compared to the long-GUT group, indicating a stronger incremental relationship between reward captures and residence times in short-GUT humans. b) Plotted is the same data for the short- (yellow) and long-GUT gerbils (red). Here, individual slopes did not differ significantly between the two groups.

In addition, the revised data analysis revealed that both species were sensitive to sunk costs. However, only humans with below-median giving-up times but all gerbils timed their patch-

leaving consistent with MVT. Lastly, a cox-regression suggested that both species used similar cues to time their patch-leaving, see revised results section p. 16-22, l. 558-753:

‘Increments in GUTs by residence times in both species

Humans tended to prolong their GUTs with increasing trial durations contradicting optimal foraging behavior (Figure 7 a)). Consistent with this notion, within-subject regressions confirmed that humans incremented their GUTs as they spent each additional second in the current patch. In both human GUT groups, we observed positive slopes for 19 out of 21 participants, [short-GUT: mean slope of = 1.020 ± 0.715 , $t(20) = 6.374$, $p < .001$; long-GUT: mean slope = 1.013 ± 1.020 , $t(20) = 4.126$, $p < .001$], suggesting that individuals in both groups consistently extended their GUTs as they spent each additional second in the current patch. There was no evidence for difference in the slope between the two human GUT groups, $p = .508$. The same pattern of results we observed also in the long- [mean slope = 1.082 ± 0.084 , $t(8) = 36.443$, $p < .001$] and short-GUT groups of gerbils [1.070 ± 0.091 , $t(8) = 33.416$, $p < .001$] (Figure 7 b).

Taken together, on average all but one gerbil’s GUT data was consistent with the predictions of a simple GUT rule. In humans, only the long-GUT group behaved in accordance with the simple GUT rule. Yet, unlike the animals, these human subjects seemed to choose suboptimal GUT durations. Given the task conditions, especially due to the quick depletion of reward, there was no benefit in prolonging residence times after the first few target encounters. Long-GUT humans who did this regardless showed significantly poorer task performance compared to short-GUT humans. This difference in performance did not exist between long- and short-GUT animals likely due to the only marginal difference in GUTs between the two groups of animals. Yet, both species showed a tendency to factor in sunk cost into their patch-leaving decision as they extended their GUTs depending on how much time they had already spent in a given patch.

Fig. 7: Testing the GUT-rule. **a)** Point plots show individual humans' GUTs and averaged ICIs in seconds (*s*) plotted for each patch quality of the long-GUT group. Connecting lines indicate the values that belong to the same individual. In all subjects GUT durations consistently exceeded the average ICIs, in accordance with a GUT rule used for patch-leaving. **b)** Corresponding GUT and ICI data shown for the short-GUT group, red lines show a downward trend indicating 14 subjects who had lower GUTs than ICIs and, thus, a GUT-ICI pattern not conforming to the GUT-rule. **c)** The point plot shows the same GUT-ICI relation for the long-GUT gerbil, the single red line indicates the deviation from the GUT-rule in a single gerbil. **d)** Point plots show short-GUT gerbils' GUT-ICI patterns. Here, all animals showed data were consistent with the GUT rule. **e)** Linear relationship between residence times and GUT durations for the human sample on group level. The green line represents the average regression line derived from within-subject regressions, predicting GUTs based on residence times within the long-GUT group. The blue line shows the same for the short-GUT group of human subjects. **f)** Linear relationship between residence times and GUT durations in gerbils. The red line equals the average regression line of all gerbils in the long-GUT group obtained from the within-subject regressions (GUT regressed on residence time), and the yellow line shows the same for the short-GUT group.

Instantaneous and average collection rates - early gerbils, belated humans

According to the marginal value theorem, optimal patch-leaving decisions are timed to the moment when the ICR approximates the MCR. As an estimate of the collection rate at which a reward capture *i* occurred, we used the inverse of the time that had passed between the previous reward capture *i-1* and the reward capture *i* (i.e., $1/\text{intertarget times}$). To approximate the collection rate in the moment of patch-leaving, we used the inverse of the observed GUT, i.e., the time since the last reward capture and leaving the current patch (McNair, 1982). The MCR we obtained by dividing the total number of reward captures by the total search time (see Wolfe, 2013). Given the difference in the number of total earnings between short- and long-GUT human subjects, we predicted that the latter group of participants had extended their residence times longer than what is considered optimal according to the MVT (i.e., estimated ICRs at the time of leaving should be significantly lower than the average collection rate).

Figure 9 a shows the trajectory of the ICRs as a function of target captures for the long-GUT human subjects and b) for the short-GUT group. Testing the MVT prediction, a one-way repeated measure ANOVA with type of time interval (ICRs for the three patch qualities as level 1-3, and the average collection rate as the level 4) yielded a significant main effect for the type of interval, $F(3, 60) = 102.502, p < .001$. In line with our prediction, post-hoc contrasts showed that all three estimated collection rates at the time of patch-leaving were below the average collection rate of 0.165 ± 0.02 rewards/s, with $p < .001$ in all three patch types. Also in the short-GUT humans we found a significant main effect of the type of time interval, Friedman $F(2.905, 58.095) = 3.457, p = .023$. However, post-hoc tests revealed that in high-quality patches, estimated ICRs at the time of patch-leaving were still significantly above the average collection rate, $p = .015$, no evidence for a difference was found in medium- and low-quality patches, with $p = .277$, and $p = .614$. Thus, at least in medium and low-quality patches we found evidence for an optimal timing of the patch-leaving according to the MVT.

Repeating the same analysis for the gerbil data, the one-way repeated measure ANOVA with the type of time interval as the single factor yielded a significant main effect in both groups of gerbils, [long-GUT: Friedman $F(2.778, 22.222) = 8.701, p < .001$; short-GUT: $F(2.778, 22.222) = 19.0, p < .001$]. Post-hoc contrasts for the long-GUT gerbils showed that in low-

quality patches the estimated collection rates at the time of leaving were still significantly higher than the MCR, $p = .001$. No evidence for such differences were found for medium- and high-quality patches, $p = .185$, $p = .670$. In short-GUT gerbils, ICRs in low- and high-quality patches were still significantly above the MCR, $p = .001$, $p = .010$, in medium-quality patches the evidence was anecdotal, $p = .052$. Figure 9 c) and d) show the trajectory of the ICRs as a function of target captures for the long-GUT and the short-GUT group of gerbils, respectively.

Fig. 9: Collection rates. Point plots show the ICRs for seven last reward captures averaged for the long-GUT (a) and short-GUT humans (b). Error bars equal \pm standard error. The gray dashed line marks the averaged overall collection rate given by the number of total rewards divided by the total search time in seconds (shaded area indicates the 95% confidence interval). The estimated capture rate at the time of leaving was defined as the inverse of the participant's GUT with short GUTs leading to higher estimated ICRs at the time of leaving compared to long GUTs. Point plots show ICRs for the long-GUT (c) and short-GUT (d) gerbils of the last three reward captures. Note that only the last three reward captures are plotted here because this was the average number of rewards obtained. In contrast to the MVT prediction, ICRs in gerbils were well above the MCR (red dashed line) at the time of leaving (i.e., the ICR at which the last reward was captured).

Formal model testing of cues used to inform patch-leaving decisions

Lastly, we used cox regressions to test different predictors that potentially increased or decreased the likelihood to stay in the current patch (e.g., Hutchinson et al., 2008). For this purpose, we used cox-regressions that model the impact of different factors on the probability to leave the current patch (i.e., the hazard ratio of patch leaving). The Cox proportional hazard model is a regression model typically used in epidemiology to find out the relationship between

the survival time of a patient and one or more predictor variables (Bender, 2009). The model has also become widely used in the foraging literature (e.g., Hutchinson et al., 2008; Lottem et al., 2018) to model the residence times using the following hazard function:

$$h(t) = h_0(t) * \exp(b_1x_1 + b_2x_2 + \dots + b_nx_n),$$

where t is the residence time, and $h(t)$ the hazard function of the residence time, b indicates the impact of the predictor x on the probability to reside in the current patch. The resulting value $\exp(b_i)$ is called the hazard ratio (HR) for the predictor i . It refers to the relative 'risk' of leaving the current patch for different levels of the i -th predictor. In other words, it quantifies the change in risk of leaving the current patch associated with a unit change in the i -th predictor. A HR greater than 1 indicates an increased risk of leaving the patch, while a hazard ratio less than 1 indicates a decreased risk. A HR of 1 indicates no change in risk.

The within-subject regressions with the number of reward captures as the predictor and the residence times as the outcome had already indicated an incremental relationship between the two variables, consistent with an incremental mechanism based on reward encounters driving patch-leaving decisions in both species. Given this finding, we entered the number of reward captures as the first predictor to the model and expected a HR below 1, i.e., a protective effect of the number of rewards decreasing the risk of patch-leaving. Since all humans and gerbils had positive slopes in the within-subject regressions, we expected this protective effect to be significant in both short- and long-GUT humans and animals. In addition, we used the averaged inter-target times between the last and the second before last reward capture, and between the second-last and the third last reward captures. This value could provide subjects with a good estimate of the recent collection rate. Increases in value should increase the risk of leaving, at least in the short-GUT group of humans that had shown collection rate data most closely in accordance with the MVT (see also Hutchinson et al., 2008). Thus, we expected a HR significantly larger than 1 in the short- but not in the long-GUT group of humans, indicating that only the former group of subjects used their recent reward capture rates as a cue for patch leaving.

In line with our prediction, the cox regression for the short-GUT humans revealed that, while controlling for the number of rewards, the hazard ratio (HR) for the averaged ICI was 5.01, 95% CI [1.30, 19.42], indicating a substantial increase in the 'risk' of patch-leaving, Wald $\chi^2(1) = 2.33$, $p = .02$. In other words, if the average ICIs of the last two target captures increased by 1 s, subjects were 5 times more likely to leave the current patch compared to no increase in the averaged ICI. Thus, consistent with the MVT, the cox model confirmed that the short-GUT humans were sensitive to declines in their current collection rates and timed their patch-leaving accordingly. Again in line with our prediction, the number of reward captures conversely appeared to be a protective factor regarding the 'risk' of patch-leaving as they decreased the risk to leave by 80% with each new reward capture, (HR = 0.20, 95% CI [0.09, 0.44], Wald $\chi^2(1) = -3.95$, $p < .005$), consistent with the incremental rule.

Next, we computed the same cox regression model also for the long-GUT group. Again, we could confirm the results of within-subject regressions that had indicated an incremental

relationship between reward captures and residence times in that the number of reward captures had a protective effect on the ‘risk’ of patch-leaving, decreasing the risk by 73%, HR = 0.27, 95% CI [0.13, 0.55], Wald $\chi^2(1) = -3.59, p < .005$. Intriguingly, in the long-GUT group, the average ICI had no effect on the ‘risk’ of patch-leaving HR = 3.89 95% CI [0.38, 39.63], Wald $\chi^2(1) = 1.15, p = .25$. Thus, in the long-GUT group, subjects’ estimates of the recent collection rate did not impact their patch-leaving. These findings are consistent with results from the collection rate data showing that long-GUT humans’ ICRs were not in agreement with the MVT.

The data obtained from gerbils with long- and short-GUT were much more consistent between the two groups. We did not find any significant differences in the effect of the number of rewards and averaged ICIs on the risk of gerbils’ patch-leaving between the two groups. Hence, the cox regression was carried out for the entire group of gerbils (pooled long- and short-GUT animals together). The results matched those of the humans in the short-GUT group: if the averaged ICI increased by one second, gerbils were almost 3.5 times more likely to leave the current patch, HR = 3.46, 95% CI [1.03, 11.69], Wald $\chi^2(1) = 2.00, p = .05$, while a new reward capture decreased the ‘risk’ of leaving by 94%, HR = 0.06, 95% CI [0, 0.90], Wald $\chi^2(1) = -2.04, p = .02$, confirming that gerbils relied on the reward encounters they experienced (incremental rule) but also on their current collection rates (MVT) in order to make patch-leaving decisions.

3) To me it seems that the authors have two options – split these studies up and perhaps collect additional data and publish them separately, or try to design tasks that could be more meaningfully compared for the two species.

Response

We would like to insist politely that the two tasks that we have chosen can be used to compare the core aspects of foraging behavior we focus on, namely the influence of changing reward contingencies on foraging decisions. Note that we do not make any claims to perceptual or motor processes which are of course very different in the two paradigms. In addition, one central objective was to design tasks that will allow us and others in the future to observe brain activity involved in exploratory foraging choices across the two species. In the human part, ongoing work analyzes fMRI data to investigate brain activity during the foraging decisions. Our design therefore mimics a patch-based foraging scenario without requiring participants to move around. From gerbils, on the other hand, we can obtain electrophysiological data, granting the possibility for animals to move around without jeopardizing the data quality. Thus, the paradigm first introduced in the rodent foraging literature developed by Lottem et al. (2018) appeared to be highly adequate for our purposes as the use of a visual search-based foraging paradigm is for humans.

4) I am also not convinced by the authors conclusion that human foraging adheres to MVT. Many studies have shown that human foraging does not adhere to MVT and the authors mention some

of those, but there are others (e.g. Hutchinson, Wilke & Todd, 2008; Wolfe, Cain and Aizelman, 2019; Gil-Gomez Muñoz-García, Pérez-Hernández & Wolfe, 2022; although the authors in the last one claim that their results actually do indicate this, I do not agree). What seems problematic to me is that in the literature, sometimes the findings on foraging are in line with MVT and sometimes they are not. It seems that the likeliest explanation for this is that MVT is not a principle that guides human foraging, in general. Instead, sometimes the fits to MVT may be roughly accurate, but with slight paradigm changes they do not fit. I cannot help feeling that there are so many factors influencing foraging that MVT will not work as a general explanatory principle for human foraging.

Response

We agree with the reviewer's point stating that the literature is somewhat inconsistent whether humans adhere to the MVT as this might depend on task specifics and change accordingly (see e.g., Wolfe, 2013). However, here we can show that under given task conditions, both gerbils and humans who did not overharvest timed their patch-leaving as predicted by the MVT. We do not claim that is necessarily the case in general. Actually, the 'overharvesting' human participants show that there are exceptions to MVT.

We would like to emphasize that our goal is to provide an analysis of reward-dependent behavior across species in our specific paradigms that can be utilized in future research for further cross-species comparisons, including their neural basis. As such, we do not claim that these behaviors will be observed universally, independent of the experimental paradigms used. This clarification is now added to the revision on **p. 2, l. 66-74**:

'Here, we report results from a behavioral study in which we tested 44 human participants in a probabilistic foraging task and compared their patch-leaving behavior to that of 18 gerbils. For this purpose, we designed two distinct foraging tasks suitable for the respective subjects. Yet, the two tasks were similar enough in their principle operationalizations to allow for comparisons between humans' and rodents' patch-leaving decisions. Our central goal was to analyze the reward-dependent foraging behavior across the two species using two specific paradigms that can eventually pave the way for more research on cross-species comparisons. This does not imply that specific patch-leaving behaviors can be expected to universally occur, independent of the experimental paradigms used.'

Furthermore we now also discuss the issue and point to previous evidence that showed deviations from the MVT in human's foraging behavior (e.g., Kristjánsson et al., 2020; Wolfe, 2013), see discussion **p. 26, l. 915-934**:

'A recent foraging study used a visual search paradigm similar to our human task and reported that human participants foraged longer in a given patch than predicted by the MVT (Kristjánsson et al., 2020). Their subjects performed either a conjunction search task (i.e., targets were defined by a combination of two features as in our experiment) or a feature search task (i.e., targets were defined by a single feature), and the results indicated similar foraging behavior for both search types. Yet, in contrast to the conjunction search task used in the

present study, the volunteers were allowed to switch between target types within a given patch. During these switches the ICR would drop well below the MCR, but the subjects stayed in the same display. This behavior is inconsistent with the MVT prediction. Additional investigations employing visual search paradigms, such as virtual berry picking experiments, have revealed deviations from the MVT's predictions under specific circumstances. These deviations are notably pronounced when patch quality exhibits significant variability, and when visual information becomes impaired to the extent that foragers are unable to discern whether a target item offers a reward (Wolfe, 2013, experiments 5 and 6). This evidence suggests that rendering foraging tasks more complex, e.g., by allowing changes between search types (Kristjánsson et al., 2020) or introducing a high degree of reward variability (Wolfe, 2013), patch-leaving behavior appears to be no longer in accordance with the MVT. In humans we showed that inter-individual differences in displaying a behavioral bias to exploit affect whether patch-leaving conforms to the MVT or not. A stronger exploitation bias leads to patch-leaving decisions that are less in keeping with the MVT in humans.'

Reviewer #2 (Remarks to the Author):

This is an interesting paper describing foraging behavior in more-or-less comparable tasks in humans and gerbils. The actual foraging task is a bit odd (not bad, just odd). After you pick an item, all the items move around and that item is, then, no longer valuable, even though it still looks the same. Some other items also lose their value. The net effect is that the probability of a good selection drops fast. The main conclusion is that humans follow a marginal value theorem (MVT) patch leaving rule, while gerbils do not. The authors argue that gerbils are using a "Giving Up Time" (GUT) rule. I think that the main issues here have to do with clarity. Here are my concerns.

Response:

Thank you for the appreciation of our paradigm. The odd aspect of item location changes was introduced to camouflage the logarithmic decline of rewarded targets (following the same function as for the gerbils) that could not be reached by simply devaluing already visited targets.

1) Did you explain why the human error bars are so much bigger than the gerbils? This is very striking in Fig 4, for example.

Response:

The substantial difference in error bar sizes between human and gerbil data, particularly evident in the original Fig 4, can be attributed to differences in the extensiveness of testing. Gerbils underwent a learning phase followed by testing over several days with repeated sessions. In contrast, human participants engaged in the experiment only in a single session. This prolonged testing period for the gerbils likely contributed to a more consistent and stable behavioral pattern across animals, ultimately leading to smaller error.

A critical factor that significantly contributes to the pronounced variability of human responses is the inherent diversity in human exploratory strategies: according to our observations, human participants varied strongly in their tendency to explore index by participants' giving-up times, despite optimal performance requiring such an approach. This variation in exploratory behavior introduced notable variation in residence times as well as total earnings times, although they all were instructed to not spend too much time per display.

Now, to address this variability in the revised analysis, we additionally included 20 behavioral datasets from participants who underwent a follow-up fMRI experiment using the same paradigm. The increased sample size allowed us to categorize participants into two distinct groups based on their giving-up times: those with extended giving-up times and those with shorter durations (median split). This novel analysis approach better accounted for the large variation in humans' propensity to explore. At the same time, it allowed us to test if subjects that differed in this propensity also differ in the rules they employed for patch-leaving. For comparison, a similar categorization strategy was applied to a likewise increased gerbil sample, although their behavior demonstrated a much higher degree of consistency across individuals. This is now added to the revised manuscript on **p.14-15, l. 491-525**:

'Splitting groups by their median giving-up times

We observed a large variation in human participants' residence times [range = 66.936 s] and giving-up times [GUTs; range = 14.419]. This was in stark contrast to the very consistent gerbil data (range of residence time = 7.866 s; range of GUT = 3.580). In human participants, residence times strongly correlated with the GUTs, $r_{\text{Pearson}} = 0.729$, $p < .001$. Like human residence times, GUTs of both the PC- and the fMRI-lab samples correlated negatively with total earnings, $r_{\text{Pearson}} = -0.668$, $p < .001$, $r_{\text{Pearson}} = -0.676$, $p = .001$. No such correlation was found in gerbils, $r_{\text{Pearson}} = 0.350$, $p = .153$

Thus, to better account for the heterogeneity in humans, we split the group by its median GUT (6.951) into a long- and a short-GUT group [long-GUT, $n = 20$: residence time: $M = 49.960 \pm 14.729$ s, $GUT M = 10.414 \pm 2.036$ s; short-GUT, $n = 20$: residence time $M = 30.415 \pm 14.729$ s, $GUT M = 4.138 \pm 1.842$ s]. Both residence times and GUTs for both subgroups are shown in Figure 6 a) and b). Unsurprisingly, the long-GUT subjects had significantly longer residence times compared to short-GUT subjects, $t(42) = 4.412$, $p < .001$. GUTs were on average significantly longer in the fMRI subjects, [fMRI: $M = 8.510 \pm 3.543$; PC: $M = 5.858 \pm 3.378$; $t(40) = 2.414$, $p = .021$. This means that more fMRI subjects entered the long-GUT group (15 out of 21 subjects), while more PC-lab participants were included in the short-GUT group (14 out of 21). Yet, importantly, within the short- and long-GUT groups there were no differences in GUTs between fMRI and PC-lab participants, [long-GUT, $p = .145$, short-GUT, $p = .150$].

For comparison, we also split the group of gerbils in the same way into a long-GUT, [GUT $M = 5.690 \pm 0.780$ s, residence time $M = 15.488 \pm 1.781$ s], and a short-GUT group [GUT $M = 4.347 \pm 0.390$, residence time $M = 11.901 \pm 1.012$ s]. Also in gerbils, residence times were significantly longer in the long-GUT group, $t(16) = 4.946$, $p < .001$, but in contrast

to humans, the two groups of gerbils did not differ in the total reward captures, $p = .722$. Figure 6 c) and d) show the gerbils' residence times as well as GUTs as a function of patch quality for both groups.

Fig. 6: Residence times and giving-up times as a function of patch-quality after group-splitting. a) Point plots show humans' residence times as a function of patch-quality. Small dots in orange indicate individual data points of the long-GUT humans, dark-red dots index individual data points of the short-GUT group. Diamonds index the mean values. b) GUTs of

humans as a function of patch-quality. c) Gerbils' individual residence times after group splitting. d) GUTs as a function of patch-quality are shown for the long- (orange) and short-GUT (dark-red) gerbils.'

2) In Fig 5C, human variability doesn't look so bad. Why is that situation different?

Response:

It was the participants' giving-up times, indexing either an exploratory or exploitative 'default' mode, in which humans differed the most. As mentioned above this led to a high variability in variables such as the residence times (see above Figure 6 a) and b)). The search behavior itself however appeared to be much more consistent. None of them took breaks while searching or

disengaged from the task, so that instantaneous collection rates (originally shown in Figure 5 c)) showed much less dispersions.

3) I worry that the high noise in the human data explains why many statistical tests are not significant. I think you said something about that but now I can't find the comment. In any case, I am worried by the noise in the human data.

Response:

see Response to 1); we increased the sample sizes in both species and revised the entire analysis performing subgroup analyzes to appreciate the great dispersion in giving-up times particularly in humans. Given the new analyzes we revised the general conclusions accordingly, see e.g., **p. 22-23, 1.773-795:**

'Discussion

This study had the goal to elaborate a specific probabilistic foraging paradigm for inter-species comparisons. To this end, we designed two separate foraging tasks that were tailored to suit each species. Although the tasks differed in their specific details, the underlying reward structure was intentionally made comparable. This allowed us to investigate both the similarities and dissimilarities in foraging behavior between animals and humans.

Humans and gerbils both adapt to changing reward probabilities.

Timing patch departures based on a fixed number of reward captures or based on a fixed amount of time only works well if the forager roams an environment that offers patches that do not differ greatly in quality (see Wilke, 2004). However, both the fixed N- as well as the fixed-T rule are not optimal if patches within an environment differ greatly in quality (Iwasa et al., 1981; Stephens & Krebs, 1987). Thus, given the randomly changing reward probabilities in our paradigms, the use of these rules would have been disadvantageous for our species. The findings that both residence times and reward captures increase with increasing patch quality, confirmed this assumption. However, humans showed high variability in individual residence times and GUTs. We therefore divided the sample of humans based on the median GUTs into long- and short-GUT individuals to evaluate whether different rules apply to these two subgroups of human participants. For better comparison we did the same also for the group of gerbils. All animal and human subjects showed evidence for the incremental rule of patch-leaving (see Figure 1 c) and were sensitive to sunk costs. Moreover, with the exception of the long-GUT humans, humans and gerbils timed their patch-leaving optimally according to the MVT. Lastly, a cox-regression suggested that both species used similar cues to time their patch-leaving.'

4) I found the discussion of the GUT rather confusing.

Response:

By revising the entire discussion, we are confident that the conclusions derived from the revised analysis are now more clear and concise, especially in those parts that were confusing or imprecise before, see p. 23-24, l. 799-865:

'Reward captures incrementally delay patch-leaving in both species

Due to the probabilistic decay function and the additionally varying initial probabilities, predicting the reward probabilities for both humans and gerbils was exceedingly challenging. To tackle single reward encounters provided a first means for our gerbils and humans to learn about the quality of the current patch and each new reward encounter could be perceived as an indication that the current patch may be of good quality, and, thus worthwhile to spend more time in it. Consistent with this notion, the within-subject regressions of residence times on the number of reward captures, showed a positive slope in all individuals of both species, demonstrating that residence times were extended incrementally following a new reward capture. This led to significantly longer residence times in high compared to low quality patches in both species and is consistent with an incremental mechanism driven by reward encounters that incrementally increase residence times by postponing the patch-leaving (Iwasa, 1981). Further support of this notion was given by the results of the cox regression in both species showing that the number of reward captures had a significant protective effect on the 'risk' of patch-leaving (i.e., delaying patch-leaving). This result confirms previous findings in humans (e.g., Wilke et al., 2009) and agrees with the results in mice by Lottem and colleagues (2018). However, in contrast to our analysis, Lottem et al. (2018) analyzed mice nose-pokes and fitted these data with a proportional hazard model. The estimated hazard rate reflected the probability to leave a current patch as a function of nose-pokes that started at its minimum with the beginning of a trial and would increase with each unrewarded nose poke. Each rewarded nose poke, however, decreased the hazard of leaving prolonging residence times. While the mice collection rates at the time of leaving were in keeping with the MVT, the incremental model showed a significantly better fit compared to a MVT-based model fitting. Using the same foraging paradigm in gerbils, but analyzing reward captures and residence times instead of nose-pokes, we can replicate this finding for gerbils. By designing a task with a foraging environment that has a comparable reward structure, we can also show that human foragers similarly adopt an incremental mechanism for patch-leaving. This demonstrates that both species adapt their foraging strategy in similar ways when facing an environment of unknown and variable reward structure in which some patches offer more rewards than others.

Overharvesting only in long-GUT humans

Although the incremental rule was evident in the data of all humans, the relationship between reward encounters and residence times was weaker (significantly shallower slopes) in the long-

compared to the short-GUT humans. Long-GUT humans showed GUT data most consistent with a simple GUT rule: their GUTs appeared to be consistent across patch-types, and always exceeded the previous ICIs. Moreover, it showed, these subjects had adopted a rather detrimental GUT rule with excessively long GUTs resulting in overharvesting (i.e., exploiting a patch longer than what is considered optimal according to the MVT). The disadvantageous timing of their patch-leaving became evident in the significantly poorer overall performance measured in terms of total reward earnings compared to short-GUT humans. In the short-GUT group of humans, in contrast, only a minority of subjects demonstrated foraging behavior with GUT rule-conforming GUT-ICI patterns. Hence, the strongest evidence for a fixed GUT rule in humans was found in subjects who had the tendency to overharvest.

There was no evidence for such a difference in performance between the two subgroups of gerbils. Both subgroups of gerbils had consistent GUTs across patch-qualities, and GUTs exceeded the ICIs in all but one animal. These findings are well in agreement with the predictions of the simple GUT rule. In accordance, also previous studies in other foraging animals facing patches of unpredictably varying quality reported patch-leaving behavior that was in agreement with a simple GUT-rule (e.g., Redhead & Tyler, 1988; Ydenberg, 1984).

A mundane interpretation for the phenomenon of overharvesting being exclusive to humans might suggest that these individuals possibly failed to understand the task. However, efforts to gauge subjects' comprehension through post-briefing questions and training performance assessment was not in accord to this hypothesis. Moreover, prior to the main experiment, all subjects were told that an exhaustive search strategy (i.e., trying to find all existing rewards per display) would be detrimental to the overall task performance. Studies in elderly foraging humans reported age-related increases in GUTs as an indication of an increased behavioral tendency to exploit (e.g., Mata et al., 2013). This suggests that exploration and exploitation as opposing behavioral tendencies together form a continuum and that individuals may differ in their position along this continuum due to age differences and other factors. For instance, a recent study in patients with opioid-use disorder showed that interindividual variability in overharvesting (in both users and controls) was related to a poorer neuromelanin signal, and indirectly catecholaminergic function (i.e., dopamine), of the VTA (Raio et al., 2022). Neurotypical subjects may already differ in their tendency to either explore or exploit based on genetic variations in those genes that control, e.g., the formation of the catecholaminergic system. Still, the extent to which long-GUT subjects tended to overharvest is striking, given the disadvantage that arose from this strategy. Clearly, more research is needed to further examine potential reasons that could explain these interindividual differences in foraging behavior.'

5) Oh....and what are the gray diamonds in Fig 4a & b.

Response:

All figures are completely revised as is the entire analysis. For more transparency and a better appreciation of raw and individual data, we did not normalize and chose dot plots showing individual data points together with the group means.

6) Maybe it would be good to have a figure illustrating the different rules that you are testing?

Response:

To improve the clarity of the manuscript and its readability, we added a new Figure 1 that summarizes all patch-leaving rules that we tested and compared between the two species, **p. 6, l. 225:**

Fig. 1. Simple heuristics to time patch-leaving decisions. Using a fixed-time rule, the patch is left independent of the number of prey encounters (green stars) (a), whereas a patch is left after a fixed number of prey encounters have been found if a fixed-number rule is used (b). According to the incremental rule, each prey capture increases the probability to stay in a patch postponing the patch-leaving (c). Using the giving-up-time rule, the tendency to stay in the patch declines as a function of unsuccessful search and each prey capture resets it to a maximum. Adopted from Wilke et al., (2009).'

7) MINOR: There are some odd phrases. In the Abstract "pertained exploitation" – what does it mean? On P4 "a fixation of these". There are some others, too.

Response:

'Pertained exploitation' referred to subjects who, instead of quickly switching between displays, continued searching unsuccessfully after the last reward capture leading to increasing giving-up times. The entire text of the manuscript has been revised, and odd phrases have been removed,
p.1 l. 30-32:

'Foraging in animals, including humans, confronts the foraging subject with the exploration-exploitation dilemma: continuous exploitation of a given patch leads to the depletion of resources, and the forager must decide when to switch location to find new energy resources.'

as well as on **p. 3, l. 125-127:**

'Similarly, in the human task, a recently collected target item not only became "inactive" and remained in the search display, but also additional target items were rendered "inactive", so that a new fixation of these deactivated targets would not result in a reward.'

8) MINOR: P11 says "in accordance with the German animal welfare law (insert here the protocol numbers)." I think you forgot to insert.

Response:

We thank the reviewer for the careful reading and have now added the protocol number. It now reads on **p. 10, l. 371-373:**

"The age of the animals during these experiments varied between three to four months. All experiments were performed in accordance with the German animal welfare law (NTP-ID: 00041189-1-X)."

Reviewer #3 (Remarks to the Author):

Summary

Foraging behavior is a fundamental necessity to animal survival across species. In this work, the authors perform a cross-species comparison of foraging strategies in humans and Gerbils. In the human version of the task, participants had to use a computer mouse to locate hidden symbolic

rewards in a virtual environment and could click to progress to the next virtual environment. In the rodent task, food-deprived gerbils had to nose-poke to receive a chance of a food pellet and could physically switch between two reward locations. Both versions of the task aim to operationalize the exploration-exploitation dilemma with a temporal cost of exploration. The underlying belief is that, since the statistics of reward availability follows similar decay dynamics, cross-species differences in the resultant behavior are therefore contrastable.

The experimental design has potential for exploring inter-species differences between exploration-exploitation strategies and the limitations of extrapolating understanding of human behavior from rodent data. The major finding of this work is that there are significant qualitative differences between the human policy for selecting a new virtual environment and the gerbil policy for moving to a new food port with replenished resources. While this observation is novel and interesting, more extensive data analyses are required to support this claim.

Major Comments

1) The authors should explain the significance of contrasting human and rodent foraging rules. From reading the introduction, it is not clear whether there are existing theories that are being tested in these experiments regarding inter-species differences, and how the results of the study change our knowledge regarding such cognitive differences.

Response:

To our knowledge there are now existing theories that elaborate on inter-species differences in foraging behavior. Key objective was to uncover potential commonalities across species in how they approach the problem to determine the optimal timing for their patch departures. Previously rodents, tested in the same behavioral task, showed patch-leaving decisions in accordance with the incremental rule of patch-leaving (Lottem et al., 2018). Single reward encounters were shown to increase trial durations incrementally. This is consistent with the idea that reward encounters likely serve as a cue that the currently visited patch of unknown quality is potentially rich of rewards and thus worth spending more time in. Thus, each novel reward encounter resets the tendency to remain in the current patch to its initial maximum at the time when the patch was entered (see Fig. 1 c) attached in response to the next major comment No. 2, Rev.3.). Even though the mice also showed patch-leaving behavior consistent with the central prediction of the MVT, model comparisons showed that a model based on the incremental rule showed a better fit to predict the behavior than a purely MVT-based model. Studies on human patch-leaving decisions provide similarly both evidence for the MVT under certain task-conditions (e.g., Wolfe 2013 experiment 1 but see experiment 5 and 6), but also good evidence for an incremental mechanism based reward encounters that incrementally extend residence times (e.g. Mata et al., 2009; Wilke et al., 2009). Our findings are in keeping with both the evidence found in the mice study as well as with the findings in humans. Importantly, the central objective was to design tasks that will allow us and others in the future to observe brain activity involved in exploratory foraging choices across the two species. In the human part, ongoing work analyzes fMRI data to investigate brain activity during the foraging decisions. Our design therefore mimics a patch-based foraging scenario without requiring participants to move around. From gerbils, on the other hand, we can obtain

electrophysiological data, granting the possibility for animals to move around without jeopardizing the data quality.

2) Also in the introduction, the authors discuss various foraging strategies and the conditions in which they could be optimal solutions to the foraging dilemma. Unfortunately, I found this important section confusing (since it appeared in an unrelated section describing the task) and lacking in rigor. I would like to see clearer, more systematic explanations of the different environment types (for example, I did not understand what is meant by “reward probabilities following a Poisson distribution”), the different patch-leaving rules, and which rule is optimal in which environment and why. This would help clarify subsequent claims, such as “Because of the unpredictable variations in the patch quality, also the GUT rule constitutes an optimal heuristic for patch-leaving decisions”.

Response:

To enhance the clarity of the introduction and the foraging strategies tested in our study, we added a new figure that depicts the different patch-leaving rules that we examined, (revised manuscript, **p. 6, l. 225**). The figure is shown in the response to reviewer 2, remark 6 on **p. 20 of this document**. In addition, we performed major revisions of large sections of the introduction to increase the clarity of our claims regarding the adequacy of different patch-leaving rules, e.g., **p 4-5, l.172-190**:

‘Similar to the incremental rule, also the giving-up time rule (GUT rule, Krebs et al., 1974; McNair, 1982, see Figure 1 d) does not require a prior judgment/knowledge about the patch-quality. The GUT rule states that a forager only tolerates a certain amount of time without a new reward capture since the last reward capture. Once this temporal threshold is exceeded, the forager leaves the patch. Each new capture, on the other hand, resets the tendency to stay. One could think of a countdown timer that starts to count down as soon as the forager enters a patch. Every reward capture resets and restarts this countdown timer. If no reward is captured before the timer expires, the forager leaves the patch. Thus, in rich patches (i.e., high reward probability), prey is encountered more frequently, and the countdown timer is reset each time so that the forager using a GUT-rule is predicted to spend on average more time in high- compared to lower-quality patches. At the same time, GUTs should be constant within individuals and should exceed the durations of individual’s intervals between two target captures (inter-target intervals; ITI): if a subject has a GUT threshold of 4 s, then the ITI should always be shorter or equal to 4 s, because the subject leaves the patch in the moment the ITI is about to exceed the 4 s threshold. Similar to the incremental rule, the GUT rule makes use of the past success rate to estimate the upcoming success rate and does not require a prior judgment of the patch quality. As a result, the GUT rule still guarantees fitness if the environment contains patches that vary widely in quality and if the patch quality is difficult to assess in advance (Wilke et al., 2009).’

We also elaborated the meaning of poisson-distributed reward probabilities on p. 3-4, l. 132-140:

‘Given a Poisson distribution, the number of preys in a patch is expected to be random and independent of the amount of time spent in the patch (Iwasa et al., 1981). In other words, the expected rate of reward on a patch does not decline over time, as it does in environments in which resources deplete as a function of time spent in the patch. Under a Poisson reward distribution, the optimal strategy is to simply spend a fixed amount of time on the patch, regardless of the number of rewards already collected, and then move on to the next patch (i.e., ‘fixed-T’ rule, see Figure 1 a). This strategy maximizes the expected rate of reward per unit time, because the average number of reward items found in a given area or time period is constant and independent of the time spent searching (see Iwasa et al., 1981).’

3) A major strength of this work is that it takes a comparative approach, studying the same computational problem in two different species. Therefore, the validity of the approach hinges on the claim that the two tasks are the same (or similar enough). However, I am not convinced that this is the case. Specifically, in the rodent version, reward probabilities decrease as a function of the number of attempts at reward in a patch, whereas in the human version the probability decreases only after rewards. This discrepancy raises the concern that the behavioral differences may be due to the difference in task structure. For example, the last inter-capture interval seems to be more informative in the human compared to the rodent task. The authors should explain why this difference is not a likely cause for the strategy differences between the two species.

Response:

The reviewer has raised a valid concern regarding a potential confound in our study, and we acknowledge this issue. Specifically, the difference in task structure, particularly the way in which reward probabilities decrease, could have influenced the observed behavioral differences between the two species. We now address this confound in the discussion and emphasize that considering the differences between the two experimental designs, commonalities in the behavioral can thereby also be understood as showing a certain robustness against typical variance factors. This section now reads as follows:

‘Comparing foraging behavior between the two species, a potential confound could have arisen from the differences in how the reward probability decreased in the two paradigms. In the gerbil task, the reward depleted with each executed nose-poke. In contrast, in the human task, the depletion followed a new reward capture but not a target fixation which would have been analogous to a nose-poke. Clearly, the reward decay following nose-pokes led to a faster depletion of the remaining reward probability. This could explain why the gerbils left patches more readily leading to very short residence times and low number of reward captures per patch compared to humans. Yet, despite these differences, there were commonalities between the species. Especially, both species consistently demonstrated an incremental mechanism driving patch-leaving. This finding agrees with that in a previous report (Lottem et al., 2018). Similarly, both species, except for the overharvesting human subjects, timed their patch-

leaving in accordance with the MVT. Importantly, the unpredictable variability of reward across patches was the same in both experiments, thus, both species had to adapt to similar environments with aggregated reward distributions. Therefore, despite the difference in how the reward decayed we are confident that the two tasks were still sufficiently similar to comparatively study patch-leaving decisions of humans and rodents. Nevertheless, future investigations should eliminate the difference in the reward decay as potential confound to further improve the alignment of the two tasks.’ (p. 27, l. 952-969)

4) In figure 4, it is not clear how the data are normalized. More generally, it is hard to get a quantitative appreciation of the behavior. I would have liked to see more raw behavioral data, for example behavior during a single session, and individual subjects’ data (the latter could be in supplementary). Also, I could not find information regarding the number of sessions it took to train and test the rodents, and how many trials were performed per session, what is meant in “the number of trials was dependent on the animals’ behavior”?

Response:

We appreciate the reviewer's comments regarding the transparency of our initial data presentation. To enhance the quantitative understanding of the behavior, we have made several improvements, including additional point plots, which allow for a clearer identification of individual values. This modification enables a more detailed examination of the data. Importantly, we included 10 more animals as well as 20 more humans in the new version of the manuscript and revised the entire data analysis to boost the informative value of the report (p. 12-22, l. 439 -773).

All animals were trained and tested in 20 consecutive days. The behavior stabilized after day 3 and thus the first three training days were discarded, see p. 10, l. 395-397:

‘After three training sessions, the animals mastered the task more quickly, and a single experimental session was typically concluded after 15-20 minutes once the animal became disengaged. Each animal performed 20 sessions on 20 consecutive days’.

A trial in the gerbil experiment was defined as the time spent foraging at one of the two spouts (trial onset was given by breaking the light barrier by entering the critical zone of the spout until releasing it by leaving the critical zone. Therefore, the number of trials depended on how often gerbils would switch between the two spouts and varied thus across animals. The reward structure (quick depletion of reward) was intended to encourage animals to leave the spout readily. But since the switches were not forced, it was still somewhat a matter of chance when animals actually switched spouts. This circumstance led to different numbers of trials per animal.

5) Related to the previous comment, behavior in such tasks is susceptible to slow changes in behavior throughout sessions (for example, an animal will likely start a session hungry and

motivated, and this motivation may decrease as it becomes sated). However, much of the analysis in this paper assumes stationary behavior (I expect that the MCR is particularly sensitive to such variations). It is therefore important to show that the behavior was indeed stationary, or account for slow behavioral fluctuations.

Response:

We have added supplementary material reporting a within-session split-half analysis of the gerbils' foraging data. Confirming the reviewer's prediction, the results indicated a decline in the animals' task-adherence. This is now mentioned in the manuscript on **p. 22, l. 754-771**:

'Restricted access to food facilitated that the animals started each daily session with appetite and motivation. However, it is important to consider that this motivation might diminish with satiation. Such a decrease in motivation could lead to reduced task commitment, potentially resulting in a higher occurrence of task-unrelated behaviors. This, in turn, could notably affect behavioral measures such as the average collection rate. To explore this hypothesis, we performed a split-half analysis within each session, comparing the behavioral parameters of gerbils between the two session segments and across the three levels of patch quality. These results indeed revealed behavioral changes consistent with a decrease in task motivation. While the number of nose pokes remained constant throughout an experimental session, increases in the inter-poke intervals, residence times, and travel times suggested an increase in the frequency or duration of task-unrelated behaviors such as grooming. However, crucial parameters describing the animals' patch-leaving behavior suggested that animals continued to optimize the timing of their patch departures despite the fading task-motivation. Although the average collection rate declined as function of session-split and GUTs increased in the second compared to first half of a session, the difference between ICRs and MCRs was on average significantly smaller in the second half. Importantly, these results do not challenge the conclusion that we made based on the results reported previously. Details of this analysis are reported in the supplementary material.'

and **Supplementary material, S1, p.1**:

'The task behavior our animals showed in the probabilistic foraging task is likely susceptible to slow and steady changes in behavior throughout a single experimental session. Although it was ensured that the animals would start a session hungry and motivated, this motivation may decrease as a function of satiation, resulting in a decreased task commitment which conversely would increase the frequency of task-unrelated behavior. The latter would particularly impact behavioral parameters such as the average collection rate. To test this hypothesis, we conducted a within-session split-half analysis and compared the gerbils' behavioral parameters between the two session parts and for each level of patch-quality. For this purpose, we split each session into a first and a second half and tested for behavioral changes between the two splits for each patch quality using a 2x3 ANOVA with session split (first half versus second half of an experimental session) and patch quality as repeated measure factors.'

The inspection of residence times showed a significant increase in the second compared to the first part of the session, as indicated by the significant main effect of session split [$F(1,16) = 23.776, p < .001$]. Also the average travel times associated with moving between the two spouts was significantly increased in the second half of a session (see Figure 1 c), [$t(17) = -3.60, p = .002$]. Yet, whereas travel and residence times increased, there was no such effect on attempted reward captures (i.e., nose pokes) [$F(1,15) = 0.080, p = .781$], the intervals between single pokes increased significantly [$F(1,16) = 73.232, p < .001$]. Altogether this pattern of results suggests a decrease of the gerbils' behavioral efficiency with increasing session duration. As animals continued foraging, each further reward capture likely contributed to their steady satiation decreasing the animals' motivation. Consequently, being less motivated to perform the task, the animal became more and more distracted and the frequency of task-irrelevant actions (e.g. grooming, sniffing etc.) increased in the second compared to the first part of a session, leading to prolonged residences and travel times as well as inter-poke intervals while the frequency of goal-directed behavior itself, i.e., the number of nose pokes remained constant. Spending more time in patches of higher quality is worthwhile and resulted in higher yields per patch in the second half of a session in medium- and high-quality patches, but it was detrimental to the average yields in low-quality patches in which it is best to spend as little time as possible (see Figure 1 c). This interaction between patch quality and session split on the number of obtained rewards [$F(2,34) = 14.271, p < .001$] showed that the increases in residence times were not realized in order to optimize the foraging, but were rather a mere product of an increasing 'task-laziness' i.e., a diminished commitment to the task in the second half of the experiment leading to more reward gains in patches of higher quality as well as to average reward losses in patches of poor quality. The diminishing task commitment consequently led to a significant drop in the average collection rate in the second compared to the first session split [$t(17) = 1.750, p = .045$].'

The MCRs were indeed affected by the animals' decreasing task motivation, but also the ICR dropped as a function of the within-session split. This resulted in MCRs and ICRs that were even more aligned in the second half of a session. This suggested that despite the decreasing task motivation, the optimality of patch-leaving as stated by the MVT was maintained regardless. This is now also reported in detail in the **Supplementary material, S1.1, p.2**:

'The decline in task commitment was also reflected in prolonged GUTs in the second half of a session (see Figure 2 a), [$F(1,17) = 48.193, p < .001$], showing that the decreases in the animals' motivation to perform the task, which led to slow non-optimal changes in the task-performance, did also affect the gerbils' patch-leaving behavior with respect to their GUTs. There was no significant effect of patch quality nor evidence for an interaction between session split and patch-quality, suggesting that the effect of the diminishing task commitment was random and affected GUTs equally independent of the the underlying patch-quality [$F(2, 34) = 1.963, p = .156, F(2, 34) = 0.801, p < .457$]. Importantly, consistent with the simple GUT rule, GUTs were constant across patch-qualities also in the second half of a session.

Given the increases in GUTs as well as the significant drop in the average collection rate (MCR) in the second compared to the first half of an experimental session (Figure 1 f), we next tested if the patch departures' alignment with the MVT changed as a function of session duration. The changes in the gerbils GUTs already suggested a further optimization of patch-leaving over time suggesting a continuous optimization of the timing of patch-leaving.

Consistent with this, ICRs at the time of leaving should be lower in the second session split in order to account for the decreased MCR, maintaining a conformity with the MVT prediction of optimal patch-leaving. ICRs at the time of leaving comparable between the two session splits would however indicate that gerbils patch-leaving would become less aligned with the MVT as a function of session duration. In line with the hypothesis that gerbils maintained an optimal timing for patch-leaving throughout the entire experimental session, ICRs at the time of leaving dropped significantly in the second session split [$F(1, 17) = 85.048, p < .001$]. Neither the main effect of patch-quality, nor the interaction between session split and patch-quality yielded statistical significance, indicating that the drop in the ICRs was comparable between patch-qualities [$F(1.421, 24.158) = 1.480, p = .244$; $F(1.835, 31.197) = 0.484, p = .605$]. Thus, accounting for the drop in the MCR, gerbils ICRs were also decreased in the second session split. Calculating the difference between ICRs and MCRs, that should be closer to zero in the case of optimal foraging, revealed a further approximation of the two rates in the second session split, indicated by a significant decrease of the ICR-MCR difference [$F(1, 17) = 60.689, p < .001$]. Again, there was no effect of patch quality, nor a significant interaction [$F(1.421, 24.158) = 1.480, p = .244$; $F(1.835, 31.197) = 0.484, p = .605$]. This finding shows that the ICRs and MCRs became more aligned as a function of session durations, consistent with optimal foraging according to the MVT.’

6) I did not understand why residence times also included travels, shouldn't they be defined as the times between the first and last nose-poke in a patch.

Response:

The residence time was indeed defined as the time between the first and the last nose-poke, whereas the gerbil travel time was given by the time consumed by moving from one spout to the other, see **p.11 | 417-419**:

‘A trial started with the first breaking of the light barrier by a nose poke lasting more than 100 ms (hit) and ended with the last release of the light barrier. This last release marked the beginning of the following travel time that ended with the first poke at the opposite spout.’

In the human task the residence time was given by the time between entering a new display and button pressing to leave the current display, see **p.8, l. 314-317**):

‘Thus, residence times (i.e., trial durations) varied and were given by the time between entering a new display and switching to the next by button press. Travel times were given by the time between the onset of the button press and the occurrence of a new display.’

Thus, in both definitions of residence time the travel time was *not* included. Only the calculation of the average collection rate took the travel time into account in accordance with Wolfe (2013), see **p.19, l. 650-652**:

‘The MCR we obtained by dividing the total number of reward captures by the total search time including travel times (see Wolfe, 2013).’

7) Why is the ICR a good measure for the instantaneous collection rate? For example, if I understand correctly a trial with two rewards separated by one second followed by five seconds of futile attempts will have the same ICR as a trial with two rewards separated by one second in which the subject left immediately after the second reward. I would expect failures to also factor in the calculation of the instantaneous reward rate. On this issue, the analysis focuses on the cost of time, however reward probability depletes with the number of pokes. It will be important to show that rodent nose-pokes occur at regular intervals or account for variable poke durations.

Response:

Indeed, we acknowledge the point about the limitation of using ICR as a measure for the instantaneous collection rate the way we did in the original manuscript. The reviewer correctly points out that our original calculation of the ICR did not account for the time spent on futile attempts or failures. Therefore we revised the calculation of the ICR at the time of leaving (i.e., estimated last ICR) and used the inverse of the foragers' giving-up time as a proxy (McNair, 1982). Given this new calculation, 'a trial with two rewards separated by one second followed by five seconds of futile attempts' does not 'have the same last ICR (at the time of leaving) as a trial with two rewards separated by one second in which the subject left immediately after the second reward.' The last ICR at the time of leaving in the first scenario is $1/5s$ whereas in the second scenario it would be $1/0s$, i.e., 1. The revised analysis of ICRs compared to MCRs are now reported on **p. 19-20, l. 644-688**. The section '*Instantaneous and average collection rates - early gerbils, belated humans*' is already reported in the response to Reviewer 1, remark 2, on **p. 8-10** in this document.

The reviewer also raises a valid point about the analysis primarily focusing on the cost of time while neglecting the effect of reward probability depletion. This is an important consideration, especially in the context of foraging behavior. However, The within-session split-half analysis of the gerbil shows indeed that poke intervals increase as a function of session duration (see supplementary material, Fig. 1 d shown below), yet the number of nose pokes remained the same (see supplementary material, Fig. 1 c shown below). This shows that the decrease of the reward probability (that depended on nose pokes) remained constant over time. The increasing inter-poke intervals together with a constant number of pokes just show that the gerbils engaged more and more in task-unrelated behavior.

Fig. 1: Split-half analysis within sessions. Residence times (a)) as well as travel times (b)) increased as a function of session split, likely due to a fading motivation to perform the task as animals got more and more satiated throughout an experimental session. Consistent with this notion, the frequency of nose pokes remained unchanged but the temporal interval between two nose pokes were significantly increased in the second session split (d)). Spending more time per patch lead to higher reward yields in medium and high-quality patches but less yield in low-quality patches (e)). Altogether, these changes in the animals' behavior led to a significant decrease in the average collection rate (f)).

Minor comments

1) "Lastly, the example stresses that humans and other animals have had highest chances to survive and pass their genes on to next generations if their attentional system evolved in such a way that it allows the constant balancing between exploitative and explorative behavior." – this seems like a non-sequitur.

Response:

We eliminated the non-sequitur.

2) "the probability of the next reward at the same spout (side of the box) decreased exponentially to zero according to a random probability." - I did not follow what it means for a probability to decay 'according to a random probability'.

Response:

To make this more clear we have now rephrased the according section as follows, **p. 3, l 123-124**):

'In the rodent paradigm, once gerbils received a reward after nose poke, the probability of the next reward at the same spout continued to decrease exponentially to zero.'

3) The authors show an equation “ $P(\text{on} = 1 \mid t_i) = A_i e^{-(n-1)/5}$ ” without defining what any of the variables are.

Response:

We added the following definitions on **p. 9, l 333-336**:

‘here t_i is the i th trial type, i.e., low-, medium-, and high-quality trials. These trial types had different exponential scaling factors $A_1 = 0.5$, $A_2 = 0.75$, $A_3 = 1$. N indicates the number of already achieved reward captures (previous target fixations that resulted in an earning) within a trial. O_n is the positive outcome of the n th target fixation (1 for reward).’